# UGPhysics: A Comprehensive Benchmark for Undergraduate Physics Reasoning with Large Language Models

Xin Xu [* 1]  Qiyun Xu [* 2]  Tong Xiao [3]  Tianhao Chen [1]  Yuchen Yan [4]  Jiaxin Zhang [1]  Shizhe Diao [5]  Can Yang [1]  Yang Wang [1]

## Abstract

Large language models (LLMs) have demonstrated remarkable capabilities in solving complex reasoning tasks, particularly in mathematics. However, the domain of physics reasoning presents unique challenges that have received significantly less attention. Existing benchmarks often fall short in evaluating LLMs' abilities on the breadth and depth of undergraduate-level physics, underscoring the need for a comprehensive evaluation. To fill this gap, we introduce UGPhysics, a large-scale and comprehensive benchmark specifically designed to evaluate **U**nder**G**raduate-level **Physics** (**UGPhysics**) reasoning with LLMs. UGPhysics includes 5,520 undergraduate-level physics problems in both English and Chinese, covering 13 subjects with seven different answer types and four distinct physics reasoning skills, all screened for data leakage. Additionally, we develop a **M**odel-**A**ssistant **R**ule-based **J**udgment (**MARJ**) pipeline specifically tailored for assessing answer correctness of physics problems, ensuring accurate evaluation. Our evaluation of 31 leading LLMs shows that the highest overall accuracy, 49.8% (achieved by OpenAI-o1-mini), emphasizes the necessity for models with stronger physics reasoning skills, beyond math abilities. We hope UGPhysics, along with MARJ, will drive future advancements in AI for physics reasoning. Codes and data are available at https://github.com/YangLabHKUST/UGPhysics.

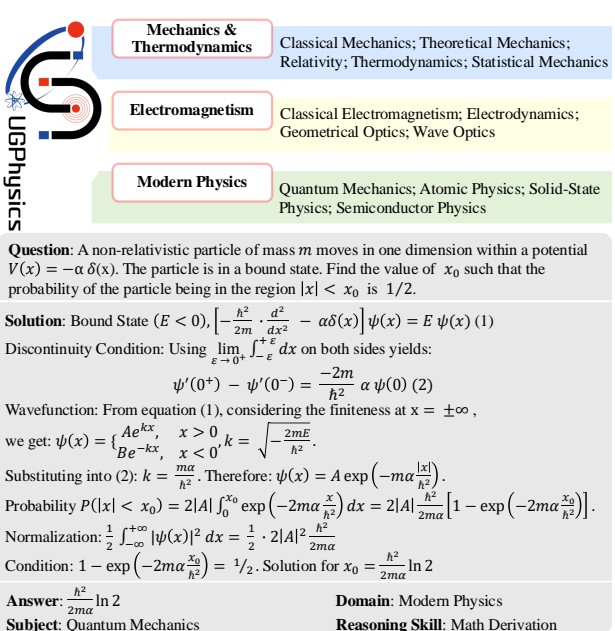

*Figure 1.* An overall illustration of UGPhysics. The top part represents the hierarchical physics domains and subjects. The bottom part showcases one concrete example.

## 1. Introduction

> *"Physics is the foundation of all the natural sciences."*
>
> — Max Planck

Physics forms the foundation for natural sciences (Planck, 1949; Hawking, 1988; Giancoli, 2000), and physics problem solving constitutes a significant aspect of reasoning for artificial intelligence (AI) (Bakhtin et al., 2019; Ding et al., 2023; Pang et al., 2024; Jaiswal et al., 2024a). Efforts to solve physics problems with machines date back to the mid-to-late 20th century (Larkin et al., 1980; Mendelson & Zelinski, 1984; Klahr & Waterman, 1986). After large language models (LLMs) have revolutionized the natural language processing community, significant attention has been paid to solving complex mathematical reasoning problems, which span several areas, such as creating challenging benchmarks (Tang et al., 2024; Gao et al., 2024; Xu et al., 2025), exploring advanced prompting techniques (Wei et al.,

---

[*]Equal contribution  [1]The Hong Kong University of Science and Technology  [2]Tsinghua University  [3]The Universify of Science and Technology of China  [4]Zhejiang University  [5]NVIDIA Research. Correspondence to: Can Yang <macyang@ust.hk>.

*Proceedings of the 42nd International Conference on Machine Learning*, Vancouver, Canada. PMLR 267, 2025. Copyright 2025 by the author(s).

2022; Wang et al., 2022), applying supervised fine-tuning (SFT) (Tong et al., 2024; Xu et al., 2024c), and leveraging continued pretraining strategies (Lewkowycz et al., 2022; Azerbayev et al., 2023). In contrast, despite its comparable, or even greater challenges for AI reasoning (He et al., 2024; Pang et al., 2024), physics has not yet garnered the same level of attention as mathematics.

Early investigations into solving physics problems were often studied alongside other scientific domains (Lewkowycz et al., 2022; Lu et al., 2022; Wang et al., 2023a). However, physics, an ancient and well-established discipline, has its unique characteristics and deserves separate treatment for AI. Unlike mathematics, which predominantly relies on logical reasoning, physics problems often require additional domain-specific knowledge (e.g., laws and principles) for resolution (Bakhtin et al., 2019; Pang et al., 2024). Moreover, physics problem-solving typically involves multiple applications of physical laws or formulas, making physics reasoning even more demanding than math reasoning. Thus, evaluating the capabilities of LLMs in physics reasoning is of significant importance. Existing physics benchmarks (Zhong et al., 2023; Zhang et al., 2023; Arora et al., 2023; Ma et al., 2024; Ding et al., 2023; Jaiswal et al., 2024b) primarily target middle or high school physics and typically are multiple-choice questions, which are relatively easy for current LLMs to handle. Although some recent benchmarks have begun to explore competition-level (He et al., 2024; Huang et al., 2024b) or college-level (Hendrycks et al., 2020; Huang et al., 2024a; Li et al., 2023) physics, they remain limited in either scope or size of undergraduate-level physics (see Table 1), which encompasses a broad range of topics and is widely used in educational assessments for humans (McDermott & Redish, 1999; Heller et al., 1992; Redish, 2003). These underscore the demand for a comprehensive benchmark specifically designed to evaluate the physics reasoning abilities of LLMs at the undergraduate level.

In this paper, we introduce **UGPhysics**, a large-scale and comprehensive benchmark tailored for evaluating the physics problem-solving abilities of LLMs across multiple **UnderGraduate-level Physics** (**UGPhysics**) disciplines, as shown in Figure 1. We carefully collect, format, split, and filter undergraduate-level physics problems (see Section 3.2), creating a benchmark comprising 5,520 distinct problems in three main domains, 13 core subjects, and 59 key topics, classified into six atomic answer types and one compound answer type. We further translate these problems into English to enable bilingual evaluation, resulting in 11,040 problems in total. To better delineate the skills needed to solve various physics problems, we categorize the problems into four distinct physics reasoning skills correlated with the requisite solution capacities (see Appendix A.3). We also conduct data leakage detection on some mainstream LLMs to validate the quality of our UGPhysics. These at-tributes are outlined more clearly in Figure 1 and Table 1. To address the challenges of answer assessment brought by unique features of physics problems (e.g., physical constants, equivalent quantities defined in problem descriptions, examples given in Table 4), we develop **M**odel-**A**ssistant **R**ule-based **J**udgment (**MARJ**) (see Section 3.3) that combines the high calculation precision of rule-based judgment methods with the flexibility of model-based approaches. Human evaluation has shown MARJ's reliability on answer judgment of physics problems (see Section 5.2).

We perform a comprehensive evaluation of 31 advanced LLMs, incorporating proprietary models, general-purpose open-source models (e.g. Qwen-2.5-Instruct (Yang et al., 2024a)), specialized math LLMs (e.g. NuminaMath (Beeching et al., 2024)), and even o1-like LLMs (e.g. DeepSeek-R1-Llama-70B (DeepSeek-AI et al., 2025)). The inclusion of math LLMs aims to assess the extent to which training on specialized math corpus contributes to physics reasoning. Despite LLMs' strong math reasoning abilities, the best overall accuracy achieved in UGPhysics is 49.8% by OpenAI-o1-mini (OpenAI, 2024b). These results highlight the challenges that UGPhysics pose to current LLMs in terms of physics problem-solving, underscoring the significance for future research with an emphasis on physics as well. To summarize our key findings:

1. UGPhysics is a challenging dataset for LLMs in physics problem-solving, with OpenAI-o1-mini achieving the highest overall accuracy of 49.8%.

2. Unlike math problem-solving, math-specialized LLMs yield only minor improvements over their general-purpose counterparts in UGPhysics, suggesting the compulsion for more high-quality physics corpora.

3. O1-like LLMs suggest a promising direction for future advancements in physics reasoning. Among them, DeepSeek-R1-Distill-Llama-70B achieves the second-highest overall accuracy on UGPhysics, though there remains a performance gap compared to top-tier closed-source LLMs.

4. Unlike abstract math reasoning, math derivation in the context of physics requires additional knowledge and involves practical meanings, where LLMs currently fall short.

5. Error analysis reveals that, unlike math reasoning, the primary types of errors made by OpenAI-o1-mini are flawed reasoning, knowledge deficiency, and wrong application.

## 2. Related Work

**Physics Benchmarks**. The growing interest in LLM evaluation has led to the creation of benchmarks across various domains, such as mathematics (Cobbe et al., 2021; Hendrycks et al., 2021; Tang et al., 2024; Liu et al., 2024; Gao et al., 2024; Xu et al., 2025), visual reasoning (Chen et al., 2021;

*Table 1.* Comparison of various benchmarks. For "Level", 1: Middle School, 2: High School, 3: College Entrance Examination, 4: Competition, 5: Undergraduate or above. "#Test" shows the number of textual test examples in physics, while "#UG" refers to the number of textual physics test examples of at least the undergraduate level. "#Subjects" specifies the number of physics subjects, and "-" means that the dataset does not divide undergraduate-level physics into more fine-grained subjects. "# Ans. Types" is the number of answer types in physics. "Language": "EN" for English and "ZH" for Chinese. "Eval." describes the evaluation methods and "Leak. Det." states whether data leakage detection is performed, which can alleviate potential test set contamination.

| Dataset | Level | # Test | # UG | # Subjects | # Ans. Types | Language | Eval. | Leak. Det. |
|---------|-------|--------|------|-----------|-------------|----------|-------|-----------|
| MMLU | 2, **5** | 548 | 118 | 3 | 2 | EN | Rule | ✗ |
| AGIEval | 3 | 200 | 0 | - | 3 | ZH | Rule | ✗ |
| C-Eval | 1, 2, **5** | 601 | 200 | - | 1 | ZH | Rule | ✗ |
| GAOKAO | 3 | 111 | 0 | - | 2 | ZH | **Rule & Model** | ✗ |
| JEEBench | 3 | 123 | 0 | - | 2 | EN | Rule | ✗ |
| CMMLU | 2, **5** | 423 | 147 | 3 | 2 | ZH | Rule | ✗ |
| TheoremQA | **5** | 131 | 131 | - | 5 | EN | Rule | ✗ |
| SciEval | - | 1,657 | - | 3 | 3 | EN | Rule | ✗ |
| PhysQA | 1 | 1,770 | 0 | 5 | - | EN | Rule | ✗ |
| GPQA | **5** | 227 | 227 | 8 | 1 | EN | Rule | ✗ |
| OlympiadBench | 4 | 376 | 0 | 5 | 4 | EN & ZH | Rule | ✓ |
| OlympicArena | 4 | 796 | 0 | 6 | **7** | EN & ZH | **Rule & Model** | ✓ |
| PhysicsQA | 2 | 370 | 0 | 6 | - | - | Rule | ✗ |
| **UGPhysics** | **5** | **11,040** | **11,040** | **13** | **7** | **EN & ZH** | **Rule & Model** | ✓ |

Cao & Xiao, 2022; He et al., 2024; Huang et al., 2024b; Lu et al., 2023), embodied AI (physical reasoning) (Bisk et al., 2019; Bakhtin et al., 2019; Wang et al., 2023b), dynamic benchmarks to mitigate test set contamination (Srivastava et al., 2024; Zhang et al., 2024; Qian et al., 2024; White et al., 2024), and many others (Chen et al., 2023; Xu et al., 2024b; Lambert et al., 2024; Mudur et al., 2024; Xu et al., 2024c; Zheng et al., 2024). Physics is an ancient yet dynamic discipline and researchers have increasingly turned to benchmarks to assess LLMs in physics reasoning. While high school-level benchmarks (Welbl et al., 2017; Lu et al., 2022; Hou et al., 2024) are valuable, they lack complex reasoning or computational challenges (He et al., 2024). College entrance exam-based benchmarks (Zhong et al., 2023; Zhang et al., 2023; Arora et al., 2023) present more advanced problems, but they often lack fine-grained subject categorization. Benchmarks like MMLU-STEM (Hendrycks et al., 2020), C-Eval-STEM (Huang et al., 2024a), and CMMLU-STEM (Li et al., 2023) include some college-level physics questions, yet they are predominantly multiple-choice questions. Advanced benchmarks such as GPQA (Rein et al., 2023), OlympiadBench (He et al., 2024), and OlympicArena (Huang et al., 2024b) provide challenging physics problems but are limited in size and breadth, often incorporating other scientific domains. Specialized physics benchmarks like PhysQA (Ding et al., 2023) and PhysicsQA (Jaiswal et al., 2024b) remain relatively simple. In contrast, our proposed UGPhysics encompasses a broader range of undergraduate-level physics subjects, offering diverse answer types, and providing a significantly larger number of test examples.

**Answer Judgment**. Evaluating model-generated answers to complex mathematical problems has long been a chal-

lenging task. Researchers have primarily relied on two approaches: rule-based methods, often combined with elaborate answer-cleaning codes (Hendrycks et al., 2021; He et al., 2024; Jaiswal et al., 2024b), and model-based methods that employ LLMs as evaluators (Gao et al., 2024). While rule-based methods are efficient, they struggle with handling complex answers (Gao et al., 2024). On the other hand, model-based methods offer more flexibility but often fall short in accurately assessing numerical values, possibly due to the current limitations of LLMs in performing precise calculations (Chen et al., 2022; Xu et al., 2025). This issue is particularly pronounced in physics, where customized relative error requirements for different problems are required. As evidenced by a 12% judgment error rate for physics problems in OlympiadBench (He et al., 2024), evaluating model-generated answers for physics problems presents an even greater challenge due to frequent appearance of physical constants and equivalent quantities (see Table 4). To address the challenge of answer assessment of physics problems, we propose MARJ, a two-stage evaluation framework that integrates both the precise calculation of rule-based judgment with the flexibility of model-based assessment (details in Section 3.3). Human evaluation has confirmed the reliability of our MARJ scoring framework .Additionally, several works (Didolkar et al., 2024; Huang et al., 2024b) utilize model-based evaluation to obtain additional metrics for assessing effectiveness.

**LLMs for Reasoning**. Significant efforts have been devoted to leveraging LLMs for solving reasoning problems, particularly in mathematics. Beyond evaluation (Liu et al., 2024; Tang et al., 2024; Gao et al., 2024), researchers have explored various approaches, including advanced prompting

Table 2. Benchmark Statistics

| Statistic | Number |
|---|---|
| Total Problems | 5520 |
| Number of Language | $\times 2$ |
| Total Domains | 3 |
| Total Subjects | 13 |
| Total Topics | 59 |
| Total Answer Types | 7 |
| Total Difficulty Level | 4 |
| Average Problem Tokens | 82.4 |
| Average Solution Tokens | 318.5 |
| Average Number of Answers | 1.34 |

Table 3. Examples of different answer types.

| Type | Abbrev. | Example |
|---|---|---|
| Numerical Value | NV | $2.51 \times 10^{-4}$ |
| Expression | EX | $\sqrt{2/\mu\sigma\omega}$ |
| Equation | EQ | $\nabla \cdot \boldsymbol{J}_\omega - i\omega\rho_\omega = 0$ |
| Interval | IN | $(-\infty, E/cB]$ |
| True/False | TF | Yes |
| Multiple Choice | MC | A |
| Compound | - | $\omega/c, \boldsymbol{k} \cdot \boldsymbol{A}_0 = 0$ |

techniques (Wei et al., 2022; Wang et al., 2022), supervised fine-tuning (SFT) (Tong et al., 2024; Xu et al., 2024c), and continued pretraining strategies (Lewkowycz et al., 2022; Azerbayev et al., 2023). To assess the impact of math-related training on physics reasoning of LLMs, we also evaluate several math-specific LLMs in UGPhysics. More recently, there has been much work dedicated specifically to physics reasoning (Pang et al., 2024; Jaiswal et al., 2024a), yet there is still a lack of a specialized physics corpus for LLMs to continually pretrain or further SFT. Our findings underscore the necessity for further research in this area.

## 3. The UGPhysics Benchmark

### 3.1. UGPhysics and MARJ Overview

We introduce UGPhysics, a large and comprehensive undergraduate-level physics benchmark specifically designed to thoroughly evaluate the physics problem-solving ability of LLMs. UGPhysics is large in size, including 5,520 physics problems presented bilingually for better evaluation. It covers three domains: Mechanics & Thermodynamics, Electromagnetism, and, Modern Physics, encompassing 13 core subjects and 59 different topics in undergraduate-level physics (details are in Appendix A.4). Similar to He et al. (2024); Huang et al. (2024b); Xu et al. (2025), each problem is structured with 7 answer types to facilitate answer judgment, including six atomic answer types and one compound type that is a list of atomic ones. To provide a more granular analysis of LLMs' physics reasoning ability, we categorize each test example into four distinct physics reasoning capabilities, which could possibly show which skill sets certain families of models succeed or fail on. Detailed statistics of UGPhysics are shown in Table 2. Additionally, data leakage detection on several LLMs is conducted to identify potential data contamination (see Section 5.4).

### 3.2. UGPhysics Creation

Our UGPhysics creation process can mainly be divided into three distinct phases: data collection & cleaning, data

processing & filtering, and problem annotation.

**Data Collection & Cleaning**. The UGPhysics is sourced from several undergraduate-level physics exercise books ("The Great Compendium of Physics Problems" from the University of Science and Technology of China, seven books in total). The corresponding PDF files are converted to LaTeX format using the Mathpix tool for optical character recognition. Both the original PDFs and the converted LaTeX files are manually reviewed and corrected by our team. The LaTeX files are then structured into a "Problem—Solution—Answer" format using various markups. Deduplication is carried out based on model embeddings to eliminate potential repeated or similar problems. Currently, problems containing images are excluded to focus on text-only reasoning of UGPhysics.

**Problem Processing & Filtering**. In physics, some problems are progressive, where subsequent questions may depend on the answers or information from previous ones. Unlike He et al. (2024); Huang et al. (2024b), we split these progressive problems into independent new problems, incorporating all relevant information in each new problem. Additionally, we exclude problems that lack definitive answers for assessing correctness, such as estimation, proof, and explanation problems. Several examples are provided in Appendix A.1. All problems are initially in Chinese and then translated into English to facilitate bilingual evaluation.

**Problem Annotation**. He et al. (2024); Huang et al. (2024b); Xu et al. (2025) suggest that classifying answer types can facilitate the evaluation pipeline. In our UG-Physics, we categorize answers into seven types: six atomic answer types and one compound type, which consists of a list of atomic answers separated by commas. One concrete example for each atomic answer type is presented in Table 3. To emphasize the focus on physics reasoning, we label each test example with one of four distinct physics reasoning skills: Knowledge Recall, Laws Application, Math Derivation, and Practical Application ("Others" for the remaining). We use GPT-4o as the annotator for the categorization of skill sets. Further details are provided in Appendix A.3.

*Table 4.* Examples of Challenges for Answer Judgment, where"GT" stands for ground-truth answers and "Model Ans." is the model-generated answers.

| GT | Model Ans. | Comments |
|---|---|---|
| 0.055 s | 55 ms | unit conversion |
| 3000 | $3.02 \times 10^6$ | intermediate precision & unit |
| $\chi_0 \frac{h\nu}{kT}$ | $\chi_0 \frac{E_2 - E_1}{kT}$ | $h\nu = E_2 - E_1$ |
| $\hbar$ | 1 | inclusion of physical constants |

### 3.3. MARJ Evaluation Framework

Evaluating model-generated answers for physics problems presents a great challenge, as evidenced by a 12% judgment error rate for physics problems in OlympiadBench (He et al., 2024). Illustrative examples are provided in Table 4. Several reasons are given as follows:

1. **Precision Issues**: The occurrence of physical constants poses challenges to the calculation precision, which could be exacerbated in multi-step reasoning, where intermediate values accumulate errors. Additionally, unit conversions or providing final answers in different units can further complicate the evaluation.

2. **Equivalent Quantities**: Physics problems often define equivalent quantities in problem descriptions, leading to multiple correct ways to express the final answer. These expressions may not always be mathematically equivalent, making it difficult to apply rule-based evaluation. Additionally, it is conventional to omit certain physical constants in the final answer, further complicating the answer judgment.

These highlight the unreliability of relying solely on traditional rule-based or model-based methods for evaluating answers to physics problems. To address these evaluation challenges, we propose a Model-Assistant Rule-based Judgment (MARJ) pipeline, which combines the efficiency of customized rule-based methods for simple answers like numerical values with the flexibility of model-based methods to handle more complex cases (see Algorithm 1).

The MARJ involves a two-stage evaluation process. In the first stage, a rule-based judgment system is employed, followed by a second stage where `GPT-4o` is used to assess cases flagged as "False" by the rule-based method. The entire pipeline is described in detail in Appendix B.4. In the rule-based matching stage, multiple answers to a given problem are evaluated individually. If any one answer deviates from the ground-truth, the result is marked as "False". For the model-based judgment stage, all answers are assessed collectively in a single evaluation prompt. For the rule-based matching stage, different types of answers are handled separately. TF and MC are judged after transforming the model-generated answers to the same format of ground-truth. For NV, answers are converted into scientific

notation and only the base of the scientific notation is considered, allowing for a relative error of up to $1e^{-2}$ to account for unit differences or rounding. EX and EQ are normalized by removing all physical constants. IN are judged by comparing the two endpoints, treating them as either NV or EX. For the model-based judgment stage, we require the evaluator model to pay attention to physical constants as well as equivalent quantities in the problem description. The few-shot judging prompt is long and will be released in our code repository. From the analysis in Section 5.2, the MARJ pipeline offers a reliable answer assessment.

## 4. Experiments

### 4.1. Experimental Setup

**Evaluated LLMs**. Our evaluation covers 31 leading LLMs, including closed-source commercial LLMs, open-source general-purpose LLMs, o1-like LLMs, and specialized math LLMs. Based on our UGPhysics, we provide a thorough evaluation of the physics reasoning capabilities of current LLMs. The evaluated LLMs are listed below:

For proprietary LLMs, we select OpenAI-o1-mini (OpenAI, 2024b), GPT4o (OpenAI, 2024a), and GPT4o-mini (OpenAI, 2024a).

For open-source general-purpose LLMs, we evaluated the LLaMA-3.1-Instruct series (8B, 70B) (Dubey et al., 2024), LLaMA-3.3-Instruct-70B, Qwen2.5-Instruct (7B, 72B)(Yang et al., 2024a), Yi-1.5-Chat (6B, 9B, 34B) (01-AI et al., 2024), Ministral-8B-Instruct-2410 (MistralAI, 2024), Mistral-Nemo-Instruct-2407 (Mistral, 2024b), Mistral-Small-Instruct-2409 (Mistral, 2024c), Mistral-Large-Instruct-2407 (Mistral, 2024a), DeepSeek-MOE-16B-Chat (Dai et al., 2024), and DeepSeek-V2-Lite-Chat (DeepSeek-AI, 2024).

We also incorporate specialized math LLMs to assess the extent to which continued training and SFT on math-related content can enhance physics reasoning: DeepSeekMath-7B (-RL, -Instruct) (Shao et al., 2024), Qwen2.5-Math (7B, 72B)(Yang et al., 2024b), Mathstral-7B (Mistral, 2023), NuminaMath-7B-CoT (Beeching et al., 2024), and OpenMath2-Llama-3.1 (8B, 70B) (Toshniwal et al., 2024).

For o1-like LLMs, we cover QwQ-32B-Preview (QwQ-Team, 2024), Skywork-o1-Open-Llama-3.1-8B (Skywork, 2024), and DeepSeek-R1 (DeepSeek-AI et al., 2025) distilled series (DeepSeek-R1-Distilled-Llama-8B, -Llama-70B; -Qwen-7B, -Qwen-32B).

We provide the details of these LLMs in Appendix B.1.

**Evaluation Setting**. Following He et al. (2024); Huang et al. (2024b), all our experiments use zero-shot prompts, tailored to different answer types for better answer extrac-

tion and rule-based matching. Detailed prompts are given in Appendix B.2. We use vLLM[1] to speed up the evaluation process. To maintain consistency in evaluations and facilitate reproduction, we set the maximum output length to 4,096 tokens and employ a greedy decoding strategy with the temperature 0. For LLMs with a maximum output length of less than 4,096 tokens during SFT, such as NuminaMath-CoT-7B, we adjust the maximum output length to align with their specific SFT configurations. More Details of the evaluation setting are given in Appendix B.3.

## 4.2. Main Results

The main results are shown in Table 5 and more detailed results are given in Appendix C. From Table 5, we have the following observations.

**Our UGPhysics presents a significant challenge for current LLMs.** The highest overall accuracy, 49.78%, is achieved by OpenAI-o1-mini, followed by DeepSeek-R1-Distill-Llama-70B with 40.17%. Notably, 15 out of 31 evaluated LLMs score below 20%, and only two models surpass the 40% overall accuracy. In contrast, OpenAI-o1-mini achieves over 90% accuracy on MATH (Hendrycks et al., 2021), over 60% on olympic and undergraduate math problems (Gao et al., 2024; Xu et al., 2025). Although current LLMs have powerful math reasoning abilities, they still struggle with complex physics reasoning.

**Open-source LLMs are catching up with closed-source LLMs, but a performance disparity remains.** Five open-source LLMs achieve overall accuracy comparable to or even exceeding GPT-4o. However, the best performing open-source LLM, DeepSeek-R1-Distill-Llama-70B, lags by around 10% behind OpenAI-o1-mini. Furthermore, GPT-4o-mini still outperforms many open-source LLMs, even surpassing some o1-like LLMs.

**Model performance improves with increasing parameter size within the same model family.** As model size grows from 7B to 72B, Qwen2.5-Math-Instruct exhibits an approximate 17% increase in overall accuracy, with consistent improvements across different domains and languages. A similar trend is observed in o1-like LLMs, where DeepSeek-R1-Distill-Llama sees an even more pronounced performance gain by 27% when scaling from 8B to 70B.

**Math-specialized LLMs outperform their general-purpose counterparts, but the improvement is less pronounced than in mathematics.** Qwen2.5-Math-7B-Instruct achieves only a 0.14% higher average accuracy than Qwen2.5-7B-Instruct, while Qwen2.5-Math-72B-Instruct outperforms Qwen2.5-72B-Instruct by 3.3%. In contrast, math-specific LLMs usually outperform their general-purpose counterparts in solving mathematical problems by a

large margin (around or over 10%) (Liu et al., 2024; Xu et al., 2025). This suggests that continued pre-training and further supervised fine-tuning on mathematical corpora yield only marginal gains in physics problem-solving, highlighting the need for future efforts to incorporate physics-specific content during training.

**O1-like LLMs yield surprisingly strong results.** DS-R1-Llama-70B achieves the second-highest overall accuracy on UGPhysics. Notably, QwQ-32B attains a competitive accuracy of 37.3%, closely approaching Qwen2.5-Math-72B-Instruct (39.5%) despite its significantly smaller model size. Furthermore, QwQ-32B outperforms DeepSeek-R1-Distill-Llama-70B on problems in ZH.

## 5. Analysis

### 5.1. Fine-grained Analysis

In this section, we conduct an in-depth analysis using 8 strong LLMs and delay the complete results to Appendix C.

**LLMs show varying performance across different subjects, although the disparity is relatively small.** As shown in Figure 2a, the average overall accuracy of eight strong LLMs reveals that they perform particularly well in Semiconductor Physics (31.0%) and Atomic Physics (26.7%). In contrast, their performance is slightly lower in Theoretical Mechanics (16.5%). Additionally, LLMs show minor performance variation across six out of 13 subjects, with accuracies hovering around 20%. In comparison, LLMs' performance can vary from 10% to 70% across different math topics (Liu et al., 2024).

**LLMs exhibit varying levels of physics reasoning skills.** As shown in Figure 2b, the selected 8 LLMs display similar performance trends across different physics reasoning skills. They perform well on Knowledge Recall tasks but struggle with Math Derivation problems. This suggests that recalling physics concepts is relatively simple for LLMs, whereas performing complex math derivations in a physics context (usually require physics knowledge and practical meanings) is more challenging. Notably, OpenAI-o1-mini outperforms the other models across all four distinct physics reasoning skills (as well as "Others").

**LLMs exhibit varying performance across different languages when solving physics problems.** From Table 5, some LLMs demonstrate only minor discrepancies in performance between English (EN) and Chinese (ZH), such as OpenAI-o1-mini, Qwen2.5-Math-Instruct, and QwQ-32B. However, other LLMs exhibit a significant performance gap between ZH and EN, such as Yi-1.5-Chat and LLaMA-3.1. For further illustration, Figure 3 presents the performance of a subset of LLMs in both languages, with the models sorted by the difference in accuracy between EN and ZH. It

---

[1]https://github.com/vllm-project/vllm

*Table 5.* **Main Results on UGPhysics** (all figures are in %). Models are classified into four different categories according to their purpose and origin. The best results within each column are **bolded** and the best results of LLMs within a similar group are underlined. "Mec. and Ther." stands for Mechanics & Thermodynamics, and "Elec." represents Electromagnetism.

| Models | Mec. and Ther. | | Elec. | | Modern Physics | | Overall | | Average |
|---|---|---|---|---|---|---|---|---|---|
| | EN | ZH | EN | ZH | EN | ZH | EN | ZH | |
| *Closed-source LLMs* | | | | | | | | | |
| OpenAI-o1-mini-2024-09-12 | **48.47** | **49.08** | **43.58** | **43.15** | **54.06** | **52.75** | **49.96** | **49.60** | **49.78** |
| GPT-4o-2024-08-06 | 36.97 | 36.84 | 36.40 | 34.58 | 42.80 | 40.63 | 39.29 | 38.01 | 38.66 |
| GPT-4o-mini-2024-07-18 | 27.81 | 26.07 | 24.84 | 22.38 | 30.72 | 29.28 | 28.51 | 26.78 | 27.64 |
| *Open-source Chat LLMs* | | | | | | | | | |
| Yi-1.5-6B-Chat | 10.99 | 7.28 | 11.99 | 7.82 | 13.66 | 9.29 | 12.26 | 8.21 | 10.24 |
| Qwen2.5-7B-Instruct | 23.89 | 20.23 | 24.09 | 17.99 | 25.57 | 21.77 | 24.62 | 20.49 | 22.55 |
| LLaMA3.1-8B-Instruct | 12.64 | 7.67 | 14.35 | 9.85 | 16.80 | 13.00 | 14.66 | 10.25 | 12.45 |
| Ministral-8B-Instruct-2410 | 13.95 | 10.81 | 15.52 | 8.99 | 19.20 | 12.17 | 16.39 | 11.07 | 13.73 |
| Yi-1.5-9B-Chat | 16.00 | 11.73 | 15.85 | 13.28 | 19.94 | 15.40 | 17.61 | 13.51 | 15.56 |
| Mistral-Nemo-Instruct-2407 | 14.08 | 11.64 | 14.78 | 11.99 | 18.41 | 16.27 | 16.00 | 13.62 | 14.81 |
| DeepSeek-MOE-16B-Chat | 3.75 | 3.18 | 4.93 | 4.82 | 7.16 | 4.49 | 5.36 | 4.00 | 4.68 |
| DeepSeek-V2-Lite-Chat | 6.50 | 4.93 | 6.75 | 5.78 | 9.47 | 7.59 | 7.77 | 6.18 | 6.97 |
| Mistral-Small-Instruct-2409 | 22.71 | 21.27 | 22.70 | 18.42 | 29.97 | 23.21 | 25.72 | 21.59 | 23.66 |
| Yi-1.5-34B-Chat | 18.79 | 13.38 | 18.63 | 12.74 | 23.17 | 17.84 | 20.58 | 15.13 | 17.85 |
| LLaMA3.1-70B-Instruct | 27.90 | 24.89 | 26.98 | 22.81 | 32.98 | 27.23 | 29.86 | 25.51 | 27.68 |
| LLaMA3.3-70B-Instruct | 33.61 | 25.81 | 33.30 | 24.84 | 39.18 | 26.83 | 35.87 | 26.07 | 30.97 |
| Qwen2.5-72B-Instruct | 35.96 | 35.70 | 33.19 | 34.05 | 37.13 | 38.22 | 35.98 | 36.47 | 36.22 |
| Mistral-Large-Instruct-2407 | 38.06 | 36.70 | 38.33 | 34.90 | 44.20 | 40.36 | 40.65 | 37.92 | 39.28 |
| *Specialized Mathematical LLMs* | | | | | | | | | |
| DeepSeek-Math-7B-Instruct | 13.25 | 12.34 | 16.49 | 13.17 | 19.07 | 15.62 | 16.21 | 13.84 | 15.03 |
| DeepSeek-Math-7B-RL | 15.17 | 11.68 | 15.20 | 12.74 | 18.54 | 15.40 | 16.58 | 13.41 | 14.99 |
| NuminaMath-7B-CoT | 13.60 | 15.52 | 14.45 | 15.52 | 18.06 | 18.50 | 15.60 | 16.76 | 16.18 |
| Mathstral-7B-v0.1 | 14.82 | 12.47 | 17.77 | 14.99 | 19.94 | 17.06 | 17.45 | 14.80 | 16.12 |
| OpenMath2-Llama-3.1-8B | 8.63 | 6.28 | 10.17 | 7.39 | 11.95 | 9.55 | 10.27 | 7.83 | 9.05 |
| Qwen2.5-Math-7B-Instruct | 23.84 | 21.05 | 22.38 | 18.09 | 26.53 | 21.34 | 24.71 | 20.67 | 22.69 |
| OpenMath2-Llama-3.1-70B | 20.31 | 18.70 | 22.16 | 18.63 | 25.44 | 21.51 | 22.75 | 19.86 | 21.30 |
| Qwen2.5-Math-72B-Instruct | 39.54 | 39.84 | 35.87 | 38.44 | 41.19 | 39.44 | 39.60 | 39.44 | 39.52 |
| *o1-like LLMs* | | | | | | | | | |
| DeepSeek-R1-Distill-Qwen-7B | 29.25 | 20.62 | 27.41 | 17.45 | 29.80 | 20.68 | 29.17 | 20.11 | 24.64 |
| Skywork-o1-Open-Llama-3.1-8B | 13.47 | 9.55 | 14.45 | 8.67 | 15.36 | 10.12 | 14.42 | 9.64 | 12.03 |
| DeepSeek-R1-Distill-Llama-8B | 16.35 | 7.15 | 17.13 | 6.53 | 20.90 | 9.08 | 18.37 | 7.84 | 13.11 |
| QwQ-32B-Preview | 36.84 | 38.01 | 35.65 | 31.58 | 38.61 | 38.92 | 37.37 | 37.30 | 37.34 |
| DeepSeek-R1-Distill-Qwen-32B | 35.22 | 28.51 | 33.40 | 23.34 | 38.09 | 28.80 | 36.11 | 27.75 | 31.93 |
| DeepSeek-R1-Distill-Llama-70B | 43.24 | 34.26 | 42.93 | 28.80 | 49.91 | 36.78 | 45.96 | 34.38 | 40.17 |
| *Newly-added Results* | | | | | | | | | |
| Phi-4 | 34.13 | 32.48 | 36.51 | 29.87 | 40.45 | 35.86 | 37.16 | 33.44 | 35.30 |
| DeepSeek-R1 | 55.49 | 56.67 | 54.50 | 48.39 | 59.90 | 57.29 | 57.16 | 55.53 | 56.34 |

is evident that LLaMA-3.3-70B-Instruct and DeepSeek-R1-Llama-70B show a substantial discrepancy between ZH and EN, while Qwen-2.5-72B-Instruct and QwQ-32B-Preview exhibit negligible differences. This discrepancy is reasonable, as LLaMA models have limited Chinese corpus for pretraining and fine-tuning (Dubey et al., 2024), whereas Qwen LLMs are trained on a much larger Chinese corpus (Yang et al., 2024a;b).

### 5.2. Reliability of Evaluation

Despite several studies utilizing LLMs to evaluate correctness across all test examples (Gao et al., 2024) or specific subsets (Zhang et al., 2023), the capability of our MARJ to reliably evaluate physics problems remains inconclusive. To substantiate our MARJ evaluation method, we conduct a human evaluation to determine its alignment with human judgment on a randomly selected subset of 100 test examples. Specifically, we initially annotate whether each solution adheres to the ground-truth answer for its corresponding problem, establishing these annotations as the gold standard. Subsequently, we compare the evaluations generated by our rule-model combination with the gold standard. We find that our MARJ evaluation achieves an accuracy of 98% when compared to human annotations, underscoring the reliability of our evaluation methods and outcomes. Furthermore, our evaluation approach is efficient in assessing correctness for examples whose answers can be easily verified by Sympy, while also demonstrating resilience in handling complex answers that are not suitable for rule-based judgments.

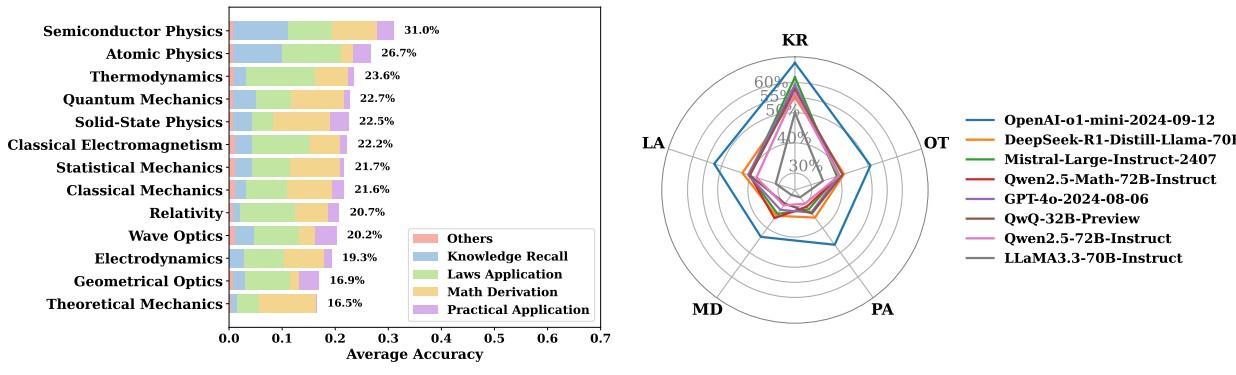

(a) Accuracy Across Subjects       (b) Accuracy Across Reasoning Skills

*Figure 2.* The distribution of overall accuracy across subjects, and physics reasoning skills. (a) The overall accuracy of different subjects averaged across 8 strong LLMs listed in Figure (b). Each bar consists of several segments with colors indicating their corresponding reasoning skills. (b) The overall accuracy of reasoning skills, averaged across all subjects. Only 8 strong LLMs are included for brevity. "KR": Knowledge Recall; "LA": Laws Application; "MD": Math Derivation; "PA": Practical Application; "OT": Others.

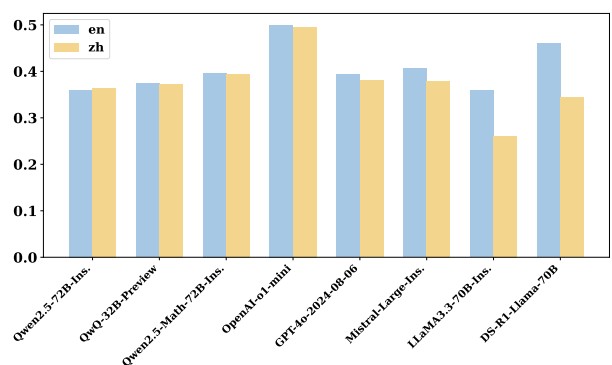

*Figure 3.* Performance in different languages, sorted by the difference of EN - ZH.

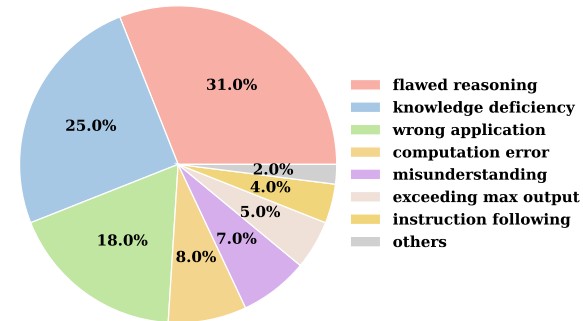

*Figure 4.* Distribution of Error Types of OpenAI-o1-mini

## 5.3. Error Analysis

To gain deeper insights into the performance of LLMs, we select 100 incorrect answers generated by OpenAI-o1-mini and have these errors annotated by human evaluators to determine failure reasons. As illustrated in Figure 4, the primary error types are flawed reasoning, knowledge deficiency, and incorrect application, which contrast with those in mathematics, where calculation is one of the major sources of errors (Chen et al., 2022; Xu et al., 2025). This suggests that reasoning and math derivation in physics, which require additional knowledge and involve real-world meanings, are more challenging than the abstract reasoning in mathematics. Several cases are provided in Appendix D.

## 5.4. About Data Leakage

We perform data leakage detection to alleviate the potential data contamination in UGPhysics. Following Xu et al. (2024a), we utilize n-gram accuracy to detect any data leak-

age within different LLMs. Concretely, we combined each problem with its solution in the dataset and randomly chose K positions for extracting 5-grams. A sample is considered contaminated if the 5-grams predicted by the model match the actual 5-grams from the dataset. The results for a subset of LLMs are presented in Table 6. It is evident that most models exhibit some degree of data leakage. Among them, Qwen2.5-MATH-72B-instruct shows the highest level of leakage, accurately predicting 5 grams in 78 samples. Additionally, we report on the contaminated samples that are subsequently answered correctly by the tested models. The numbers of both "Contaminated" and "Contaminated & Correct" samples are extremely low, suggesting that data leakage has minimal impact on UGPhysics.

## 5.5. Compare to Existing Benchmarks

**Accuracy Compared to Existing Benchmarks** To contextualize our UGPhysics's difficulty, we evaluate `GPT-4o`

*Table 6.* The proportion (in %) of data leakage detection for: (a) The proportion of contaminated examples. (b) The proportion of contaminated and correct examples. "Prop." stands for proportion.

| Model | (a) Prop. | (b) Prop. |
|---|---|---|
| DeepSeek-Math-7B-RL | 0.0% | 0.0% |
| LLaMA3.1-8B-Instruct | 0.53% | 0.06% |
| LLaMA3.3-70B-Instruct | 0.65% | 0.29% |
| Qwen2.5-Math-7B-Instruct | 0.65% | 0.36% |
| Qwen2.5-Math-72B-Instruct | 0.75% | 0.68% |
| QwQ-32B-Preview | 0.71% | 0.65% |
| DeepSeek-R1-Distill-Qwen-7B | 0.0% | 0.0% |
| DeepSeek-R1-Distill-Qwen-32B | 0.0% | 0.0% |

on established physics reasoning datasets (Table 7). GPT-4o achieves 38.67% accuracy, weaker than its 53.60% on GPQA (Rein et al., 2023), highlighting UGPhysics's greater challenge. We also include MATH (Hendrycks et al., 2021) for reference.

**Tokens Spent on UGPhysics**  We further analyze the token usage required to solve problems in our benchmark. DeepSeek-R1 (DeepSeek-AI et al., 2025), when used without maximum token constraints (via API), requires an average of 5,555 tokens to solve problems in our benchmark. To assess the impact of token restrictions, we set a maximum output token limit of 8,192 for DeepSeek-R1-Distill-Qwen-32B in our experimental setup. Under this constraint, the model requires 3,079 tokens on the MATH (Hendrycks et al., 2021) benchmark and 4,081 tokens on our UGPhysics. These results highlight the computational cost of solving problems in our benchmark relative to other tasks.

*Table 7.* Performance on various benchmarks.

| Dataset | Performance (%) |
|---|---|
| MMLU (college physics) | 68.60 |
| MMLU (high school physics) | 72.80 |
| MMLU (conceptual physics) | 92.30 |
| MMLU-pro | 75.06 |
| OlympicArena | 55.92 |
| GPQA | 53.60 |
| MATH | 76.60 |
| UGPhysics | **38.67** |

### 5.6. About the Clip Ratio

The maximum generation token limit was set to 8,192 in our evaluation. Open-source o1-like LLMs typically consume more tokens than OpenAI's o1-mini when solving UG-Physics problems. With a 8,192 token limit, only 2% of o1-mini's generations exceed this limit, whereas significantly higher proportions of other open-source o1-like LLMs fail to terminate within the limit (Table 8). To assess whether increasing the token limit improves performance, we extended

the limit to 16,384. Results in Table 9 show marginal performance gains, with a small proportion of generations still exceeding the extended limit. These findings highlight the need to address token redundancy in o1-like LLMs during reasoning, as noted by Chen et al. (2024).

*Table 8.* The Clip Ratio (in %) for Different o1-like LLMs.

| Models | 8192 | 16382 |
|---|---|---|
| DeepSeek-R1-Distill-Qwen-7B | 44.40 | 38.90 |
| DeepSeek-R1-Distill-Qwen-32B | 38.55 | 34.47 |
| DeepSeek-R1-Distill-Llama-8B | 52.37 | 43.80 |
| DeepSeek-R1-Distill-Llama-70B | 19.16 | 12.25 |
| QwQ-32B-Preview | 19.01 | 8.54 |
| o1-mini-2024-09-12 | **2.01** | - |

*Table 9.* Performance (in %) of Different Token Budget.

| Models | 8192 | 16382 |
|---|---|---|
| DeepSeek-R1-Distill-Qwen-7B | 24.64 | 24.86 |
| DeepSeek-R1-Distill-Qwen-32B | 31.93 | 32.21 |
| DeepSeek-R1-Distill-Llama-8B | 13.11 | 14.51 |
| DeepSeek-R1-Distill-Llama-70B | 40.17 | 41.77 |
| QwQ-32B-Preview | 37.34 | 38.90 |

## 6. Conclusion

In this study, we propose UGPhysics, a comprehensive undergraduate-level physics benchmark, and the MARJ answer scoring framework to evaluate LLMs' capabilities in solving physics problems. Through our extensive experiments, we find that although current LLMs excel in mathematical reasoning, there remains considerable potential for improvement in their performance on physics problems. We believe our dataset and codes will be valuable for the future development of AI with strong physics reasoning abilities.

## Impact Statement

This paper introduces an undergraduate-level physics benchmark aimed at advancing AI capabilities in physics problem-solving. Future directions include incorporating problems with images to enable multi-modal evaluation or more language to facilitate multi-lingual assessment.

## Acknowledgments

This work was partially supported by a grant from the Research Grants Council of the Hong Kong Special Administrative Region, China (Project Reference Number: AoE/E-601/24-N). We would like to thank all the reviewers for their valuable suggestions to improve our work.

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

# A. Detailed Statistics of UGPhysics

## A.1. Problem Filtering

During problem processing, we find some physics problems do not have definite answers for evaluation. Two examples are given in Table 10. We exclude these problems for automatic evaluation in our UGPhysics.

## A.2. Answer Types

By carefully reviewing a large collection of problems and referring to various past benchmarks (He et al., 2024; Huang et al., 2024b; Xu et al., 2025), we classify all answers to seven categories in total: six atomic and one compound type. Each compound type is composed of a list of atomic ones. These types are designed to encompass a wide range of problems. Detailed definitions for each answer type can be found in Table 11. Similar to He et al. (2024), the compound answer type is characterized by the attribute is_multiple_answers.

## A.3. Physics Reasoning Skills Annotation

The definitions of different physics skills are as follows:

- **Knowledge Recall**: Refers to the recall and understanding of basic physics concepts, formulas, and definitions. This category assesses the ability to accurately remember key physics points, such as the units, definitions, and fundamental properties of physical quantities, without requiring deep reasoning or complex calculations.

- **Laws Application**: Involves understanding and applying important physical laws. Students are expected to correctly describe the content, conditions, and scope of these laws and determine their applicability to specific scenarios. Examples include Newton's laws, the laws of thermodynamics, and Coulomb's law.

- **Math Derivation**: Focuses on the mathematical derivation and logical proof of physics formulas or principles. This requires students to use known laws or principles and apply rigorous mathematical reasoning to derive new relationships or theorems. Examples include deriving the momentum conservation equation from Newton's second law or deriving thermodynamic relations from the ideal gas law.

- **Practical Application**: Emphasizes the practical use of physics knowledge, laws, and derivations to solve real-world problems. This includes analyzing scenarios, building physical models, and using calculations and logical reasoning to arrive at solutions. Examples include calculating the final velocity of a car using the work-energy theorem or analyzing the motion of a charged particle using electric field equations.

The prompt used for classification is given in Table 12. Note that there are problems that fail to be classified into certain types of skill, we annotate them as "others".

## A.4. Distribution of Problems

Our UGPhysics include 3 key domains and 13 core subjects in undergraduate-level physics. The detailed topics across different subjects and the corresponding number of examples are listed in Table 13. There are 59 topics in total. We illustrate the words of topics in Figure 5. Furthermore, the distribution information of our benchmark on different subjects and reasoning skills is presented in Table 14.

*Table 10.* Estimation and Explanation Problem Example.

| Estimation Problem |
|---|
| Provide the best estimate of the cosmic ray flux at sea level. |
| **Explanation Problem** |
| Water is a polar molecule. Discuss the effect of electrode polarization on its dielectric constant at high and low frequencies. |

# B. Detailed Experimental Setup

## B.1. Evaluated LLMs

Our evaluation encompasses a range of LLMs, including both proprietary commercial models and publicly accessible models. For open-source LLMs, we cover general-purpose LLMs, o1-like LLMs, and specialized math LLMs.

Closed-source LLMs are listed as follows:

- **o1-preview** (OpenAI, 2024b): An early preview of OpenAI's o1 model, designed to reason about hard problems using broad general knowledge about the world. We used `o1-preview-2024-09-12` for our evaluation.

- **GPT-4o** (OpenAI, 2024a): GPT-4o is a multimodal LLM, and has the same high intelligence as GPT-4 Turbo but is much more efficient. For evaluation, we use this specific version: `GPT-4o-2024-08-06`.

- **GPT-4o-mini**: GPT-4o-mini is even more efficient and cheaper than GPT-4o with the cost of minor performance drop. We use `GPT-4o-mini-2024-07-18` for our experiments.

The following open-source general-purpose LLMs are evaluated on our UGPhysics:

- **LLaMA-3.1-Instruct** (Dubey et al., 2024): LLaMA-3.1 models are the most capable of the LLaMA families as of writing this paper. We used instruction finetuned 8B and 70B versions of the model. These models are licensed under Llama 3.1 Community License.

- **Ministral-8B-Instruct-2410** (MistralAI, 2024): mrl License

- **Mistral-Nemo-Instruct-2407** (Mistral, 2024b): Apache 2.0

- **Mistral-Small-Instruct-2409** (Mistral, 2024c): MRL License.

- **Mistral-Large-Instruct-2407** (Mistral, 2024a): MRL License.

- **Qwen2.5-Instruct** (Yang et al., 2024a): Qwen2.5 series are developed with dedication to math and coding. We used 7B and 72B models. 7B models are licensed under Apache 2.0, while 72B models are under Qwen License.

- **Yi-1.5-Chat** (01-AI et al., 2024): Yi-1.5 delivers stronger performance in coding, math, reasoning, and instruction-following capability compared to its predecessor. We used 6B, 9B, 34B variants. Yi-1.5 series are licensed under Apache 2.0.

- **DeepSeek-V2-Lite-Chat** (DeepSeek-AI, 2024): model under Model License code under MIT License.

- **deepseek-moe-16b-chat** (Dai et al., 2024): Model under Model License, code under MIT License.

For o1-like LLMs, we use the following:

- **QwQ-32B-Preview** (QwQ-Team, 2024): QwQ-32B-Preview is developed by the Qwen Team, focused on advancing AI reasoning capabilities. It is under Apache 2.0 License.

*Table 11.* Descriptions of answer types included in evaluation prompts, only include English version for simplicity.

| Answer Type | English Answer Type Description |
| --- | --- |
| NV | a numerical value without units |
| EX | an expression |
| EQ | an equation |
| INT | a range interval |
| TF | either True or False |
| MC | one option of a multiple choice question |

- **Skywork-o1-Open-Llama-3.1-8B** (Skywork, 2024): Skywork-o1-Open-Llama-3.1-8B is an LLM that incorporates o1-like slow thinking and reasoning capabilities. It is developed by the Skywork team.

- **DeepSeek-R1 Distilled serires** (DeepSeek-AI et al., 2025): These LLMs are distilled from DeepSeek's first-generation reasoning models DeepSeek-R1. Model under Model License, code under MIT License. In our evaluation, we use DeepSeek-R1-Distilled-Llama-8B, -Llama-70B; -Qwen-7B, -Qwen-32B.

We also experiment with the following specialized math LLMs in our study:

- **DeepSeekMath-7B** (Shao et al., 2024): DeepSeekMath is initialized with DeepSeek-Coder-v1.5 7B and continues pre-training on math-related tokens. We tested both DeepSeekMath-7B-RL and DeepSeekMath-7B-Instruct variants. Models are under Model License while code is under MIT License.

- **Qwen2.5-Math** (Yang et al., 2024b): Qwen2.5-Math is a series of specialized math language models built upon the Qwen2.5 LLMs. We evaluated 7B and 72B variants. They are under the same license as the Qwen2.5-Instruct series.

- **Mathstral-7B** (Mistral, 2023): Mathstral stands on the shoulders of Mistral 7B and specializes in STEM subjects. This model is published under Apache 2.0.

- **Numinamath-7B-CoT** (Beeching et al., 2024): This model is finetuned from DeepSeekMath-7B-base with two stages of supervised fine-tuning to solve math problems using chain of thought (CoT). It is licensed under Apache 2.0.

- **OpenMath2-Llama-3.1** (Toshniwal et al., 2024): These are specialized math LLMs that have undergone SFT on 2.3M augmented GSM-8K and MATH training examples. These models are trained by Nvidia and are licensed under Llama 3.1 Community License.

## B.2. Evaluation Prompts

The prompts employed in our experiments are presented in Table 15, with detailed explanations of response types available in Table 11. For simplicity, we here only show prompts used by English problems. The prompts for Chinese problems are quite similar and will be released in our code repository. Based on He et al. (2024); Huang et al. (2024b), these prompts are tailored for diverse subjects and response types to improve evaluation efficiency. It should be noted that for chat models, we will adhere to their official chat template.

## B.3. Evaluation Parameters

To maintain consistency in evaluations and facilitate reproduction, we set the maximum output length to 4,096 tokens and employ a greedy decoding strategy with the temperature 0. For LLMs with a maximum output length of less than 4,096 tokens during SFT, such as NuminaMath-CoT-7B, we adjust the maximum output length to align with their specific SFT configurations. For `OpenAI-o1-mini`, we set it to 8,192 tokens, as this model often requires a higher token count for reasoning tasks. Exceeding the maximum output length can result in no output being returned. Similarly, we also set the maximum output length o1-like LLMs to be 8,192 tokens. The temperature for the `OpenAI-o1-mini` model is restricted to a value of 1 due to new regulations from OpenAI. Setting the temperature to other values would result in an error from the API.

## B.4. MARJ Details

The whole pipeline of MARJ is given in Algorithm 1. For the first stage, different types of answers (TF, MC, NV, EX, EQ, IN) are handled as follows:

- For TF and MC, answers are judged after transforming model-generated answers to the same format as the golden answers.

- For NV, answers are converted into scientific notation. Only the bases of the scientific notation are considered, allowing for a relative error of up to to account for unit differences or rounding.

- For EX and EQ, answers are normalized by removing all physical constants. The physical constants are listed in Table 16.

- For IN, answers are judged by comparing the two endpoints, treating them as either NV or EX based on the context.

In the second stage, `GPT-4o` is employed to evaluate answers that were flagged as "False" by the rule-based system. We manually design judging prompts based on the prompt provided by Gao et al. (2024). As this few-shot prompt is long, we will release it in our code repository.

## C. More Results

Results across different subjects are given in Table 17, 18 and 19. Results across different physics reasoning skills are provided in Table 20.

## D. Error Analysis

We perform error analysis in Section 5.3, and here we showcase several examples of various error types in Table 21, 22, and 23.

*Table 12.* The prompt for annotating physics reasoning skills, where problem, solution, and answer represent the components of the problem to be annotated.

Problem:
{problem}

Solution:
{solution}

Answer:
{answer}

Classification Categories:
1. Knowledge Recall: Refers to the recall and understanding of basic physics concepts, formulas, and definitions. This category assesses the ability to accurately remember key physics points, such as the units, definitions, and fundamental properties of physical quantities, without requiring deep reasoning or complex calculations.
2. Laws Application: Involves understanding and applying important physical laws. Students are expected to correctly describe the content, conditions, and scope of these laws and determine their applicability to specific scenarios. Examples include Newton's laws, the laws of thermodynamics, and Coulomb's law.
3. Math Derivation: Focuses on the mathematical derivation and logical proof of physics formulas or principles. This requires students to use known laws or principles and apply rigorous mathematical reasoning to derive new relationships or theorems. Examples include deriving the momentum conservation equation from Newton's second law or deriving thermodynamic relations from the ideal gas law.
4. Practical Application: Emphasizes the practical use of physics knowledge, laws, and derivations to solve real-world problems. This includes analyzing scenarios, building physical models, and using calculations and logical reasoning to arrive at solutions. Examples include calculating the final velocity of a car using the work-energy theorem or analyzing the motion of a charged particle using electric field equations.

Instructions for Classification: Please classify the above problem by selecting the most appropriate category that best represents the type of thinking and approach required to address the physics problem. Consider the complexity, the need for creativity, and the depth of knowledge required. You should conclude your response with Šo, the problem can be categorized as ANSWER.; where ANSWER should be one of the indexes in 1, 2, 3, 4.

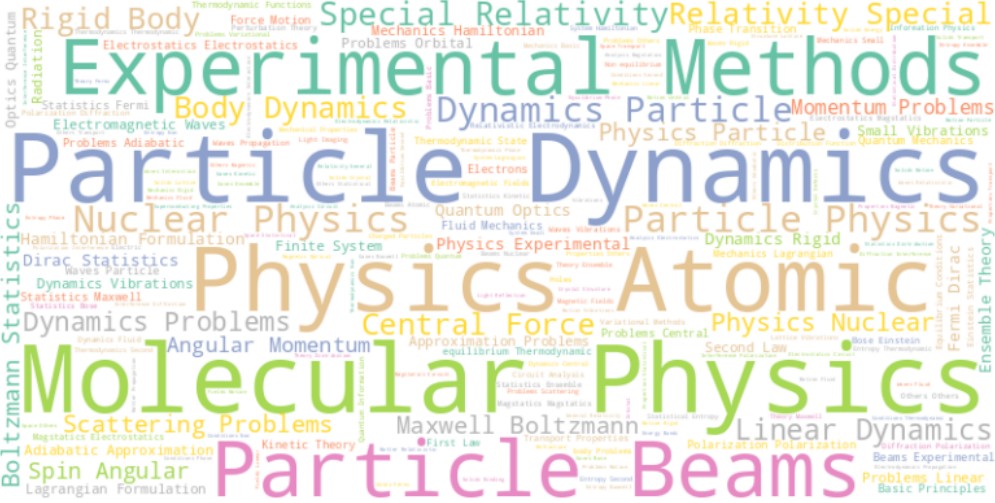

*Figure 5.* Word Cloud of Topics in UGPhysics

| Subject | Topics | # Examples |
|---|---|---|
| Classical Mechanics | Particle Dynamics | 644 |
| Classical Mechanics | Vibrations and Waves | 358 |
| Classical Mechanics | Central Force Motion | 156 |
| Classical Mechanics | Rigid-Body Dynamics | 360 |
| Classical Mechanics | Fluid Mechanics | 154 |
| Theoretical Mechanics | Lagrangian Formulation of Mechanics | 202 |
| Theoretical Mechanics | Small Vibrations of Finite System | 182 |
| Theoretical Mechanics | Hamiltonian Formulation of Mechanics | 254 |
| Relativity | General Relativity | 36 |
| Relativity | Special Relativity | 368 |
| Relativity | Relativistic Cosmology | 10 |
| Thermodynamics | Phase Transition and Equilibrium | 168 |
| Thermodynamics | Thermodynamic State and First Law of Thermodynamics | 146 |
| Thermodynamics | Second Law of Thermodynamics and Entropy | 218 |
| Thermodynamics | Thermodynamic Functions and Equilibrium Conditions | 134 |
| Thermodynamics | Non-equilibrium Thermodynamics | 78 |
| Statistical Mechanics | Ensemble Theory | 234 |
| Statistical Mechanics | Maxwell-Boltzmann Statistics | 294 |
| Statistical Mechanics | Distribution Function and Statistical Entropy | 68 |
| Statistical Mechanics | Fermi-Dirac Statistics | 228 |
| Statistical Mechanics | Bose-Einstein Statistics | 140 |
| Statistical Mechanics | Kinetic Theory of Gases | 156 |
| Classical Electromagnetism | Magstatics | 296 |
| Classical Electromagnetism | Electrostatics | 368 |
| Classical Electromagnetism | Circuit Analysis | 116 |
| Electrodynamics | Propagation and Radiation of Electromagnetic Waves | 194 |
| Electrodynamics | Relativistic Electrodynamics | 116 |
| Electrodynamics | Interaction of Electromagnetic Fields with Matter | 58 |
| Geometrical Optics | Imaging of Light | 88 |
| Geometrical Optics | Reflection and Refraction of Light | 28 |
| Wave Optics | Diffraction | 192 |
| Wave Optics | Interference | 110 |
| Wave Optics | Polarization | 302 |
| Quantum Mechanics | Linear Dynamics Problems | 332 |
| Quantum Mechanics | Basic Principles of Quantum Mechanics | 180 |
| Quantum Mechanics | Central Force and Scattering Problems | 192 |
| Quantum Mechanics | Orbital and Spin Angular Momentum Problems | 270 |
| Quantum Mechanics | Motion of Charged Particles in Electric and Magnetic Fields | 60 |
| Quantum Mechanics | Adiabatic Approximation Problems | 224 |
| Quantum Mechanics | Scattering Problems | 110 |
| Quantum Mechanics | Variational Methods and Perturbation Theory | 120 |
| Quantum Mechanics | Quantum Information Physics | 96 |
| Quantum Mechanics | Few-body Problems | 104 |
| Quantum Mechanics | Others | 104 |
| Quantum Mechanics | Quantum Optics | 246 |
| Atomic Physics | Atomic and Molecular Physics | 622 |
| Atomic Physics | Particle Physics | 394 |
| Atomic Physics | Nuclear Physics | 400 |
| Atomic Physics | Experimental Methods and Particle Beams | 414 |
| Solid-State Physics | Crystal Structure | 38 |
| Solid-State Physics | Lattice Vibrations and Mechanical Properties of Solids | 84 |
| Solid-State Physics | Crystal Defects and Motion | 34 |
| Solid-State Physics | Binding in Solids | 54 |
| Solid-State Physics | Transport Properties of Electrons and Holes in Solids | 60 |
| Solid-State Physics | Motion of Electrons in Electromagnetic Fields | 42 |
| Solid-State Physics | Energy Bands in Solids | 32 |
| Semiconductor Physics | Magnetic, Optical, and Superconducting Properties | 50 |
| Semiconductor Physics | Transport Properties | 80 |
| Semiconductor Physics | Others | 186 |
| Semiconductor Physics | Statistical Distribution of Electrons and Holes in Space | 56 |

*Table 13.* Topics of each subject and corresponding number of examples included in UGPhysics.

| | Knowledge Recall | Laws Application | Math Derivation | Practical Application | Others | All |
|---|---|---|---|---|---|---|
| Classical Mechanics | 72 | 582 | 774 | 172 | 72 | 1,672 |
| Theoretical Mechanics | 30 | 124 | 448 | 20 | 16 | 638 |
| Relativity | 10 | 188 | 154 | 44 | 18 | 414 |
| Thermodynamics | 36 | 388 | 246 | 46 | 28 | 744 |
| Statistical Mechanics | 112 | 322 | 590 | 38 | 58 | 1,120 |
| Classical Electromagnetism | 60 | 338 | 314 | 44 | 24 | 780 |
| Electrodynamics | 28 | 142 | 156 | 38 | 4 | 368 |
| Geometrical Optics | 8 | 48 | 26 | 30 | 4 | 116 |
| Wave Optics | 52 | 236 | 130 | 160 | 26 | 604 |
| Quantum Mechanics | 226 | 584 | 1,052 | 100 | 76 | 2,038 |
| Atomic Physics | 422 | 784 | 282 | 282 | 60 | 1,830 |
| Solid-State Physics | 36 | 66 | 158 | 70 | 14 | 344 |
| Semiconductor Physics | 76 | 112 | 110 | 62 | 12 | 372 |
| Grand Total | 1,168 | 3,914 | 4,440 | 1,106 | 412 | 11,040 |

*Table 14.* Statistics of UGPhysics across different subjects and physics reasoning skills.

*Table 15.* Evaluation prompts for English problems with single answers or multiple answers. {problem} is the specific problem to evaluate. {subject} denotes the subject this problem belongs to and all subjects are given in Figure 1. {answer_type_description} are specified in Table 11.

---

Evaluation Prompt for Single Answer

---

The following is an open-ended problem from {subject} of undergraduate-level Physics.
The answer of The problem should be {answer_type_description}.
Please calculate the answer according to the given requirements and the information provided.
Please use LaTeX format to represent the variables and formulas used in the solution process and results.
Please end your solution with "So the final answer is ⌐answer⌐ (unit)." and give the result explicitly, note that the unit of the answers should not be included in ⌐.

{problem}

---

Evaluation Prompt for Multiple Answers

---

The following is an open-ended problem from {subject} of undergraduate-level Physics.
The question has multiple answers, with the answers in order being {answer_type_description}.
Please calculate the answer according to the given requirements and the information provided.
Please use LaTeX format to represent the variables and formulas used in the solution process and results.
Please end your solution with "So the final answer is ⌐multiple answers connected with commas⌐." and give the result explicitly, note that the unit of the answers should not be included in ⌐.

{problem}

---

**Algorithm 1** MARJ Pipeline

---

**Input:** Problem $P$, Solution $S$, Golden Answer List $GT$, Model Solution $s$, Model Answer List $A$.
Initialize $Correctness = False$
**if** $len(A)$ not equals $len(GT)$ **then**
   return $False$
**end if**
**for** $gt, a$ in $GT, A$ **do**
   $flag = False$
   **if** $gt$ equals $a$ **then**
      $flag = True$; continue
   **end if**
   **if** $P$ is a T/F or MC question **then**
      Transform $gt, a$ to standard forms: $gt', a'$
      **if** $gt'$ equals $a'$ **then**
         $flag = True$; continue
      **end if**
   **end if**
   **if** $gt$ is expression or equation **then**
      Normalizing $a, gt$ to $a', gt'$ by removing physical constants
      **if** $gt$ equals $a$ as equation or $gt'$ equals $a'$ as equation **then**
         $flag = True$; continue
      **else if** $gt$ equals $a$ as expression or $gt'$ equals $a'$ as expression **then**
         $flag = True$; continue
      **end if**
   **end if**
   **if** $gt$ is Numeric Value **then**
      Transform $a, gt$ into scientific notation: $a = a_{base} \times 10^{a_{exp}}$, $gt = gt_{base} \times 10^{gt_{exp}}$
      **if** $|a_{base} - gt_{base}|/|gt_{base}| < \epsilon$ **then**
         $flag = True$; continue
      **end if**
   **end if**
   **if** $gt$ is interval **then**
      let $c, d$ be endpoints of $gt$; $e, f$ be endpoints of $a$
      **if** $c$ equals $e$ as NV or EX and $d$ equals $f$ as NV or EX **then**
         $flag = True$; continue
      **end if**
   **end if**
**end for**
**if** $flag$ equals $True$ **then**
   return $True$
**else**
   return $ModelJudge(P, S, GT, s, A)$
**end if**

---

*Table 16.* Physical Constants in MARJ.

| Physical Quantity | Symbol |
|---|---|
| Speed of light in vacuum | $c$ |
| Newtonian constant of gravitation | $G$ |
| Avogadro constant | $N_A$ |
| Universal gas constant | $R$ |
| Boltzmann constant (macroscopic) | $R/N_A$ |
| Molar volume of ideal gas | $V_m$ |
| Elementary charge (proton charge) | $e$ |
| Electron mass | $m_e$ |
| Electron charge-to-mass ratio | $-e/m_e$ |
| Proton mass | $m_p$ |
| Neutron mass | $m_n$ |
| Vacuum permittivity (electric constant) | $\varepsilon_0$ |
| Vacuum permeability (magnetic constant) | $\mu_0$ |
| Electron magnetic moment | $\mu_e$ |
| Proton magnetic moment | $\mu_p$ |
| Bohr radius | $a_0$ |
| Bohr magneton | $\mu_B$ |
| Nuclear magneton | $\mu_N$ |
| Planck constant | $\hbar$ |
| Planck constant | $h$ |
| Fine-structure constant | $\alpha$ |
| Rydberg constant | $R_\infty$ |
| Compton wavelength | $\frac{\hbar}{mc}$ |
| Proton-electron mass ratio | $\frac{m_p}{m_e}$ |
| Boltzmann constant | $k$ |

*Table 17.* **Results across subjects of Mechanics & Thermodynamics on UGPhysics** (all figures are in %). Models are classified into four different categories according to their purpose and origin. The best results within each column are **bolded** and the best results of LLMs within a similar group are underlined.

| Models | Class. Mech. | | Theor. Mech. | | Relativity | | Ther.dyn. | | Stat. Mech. | |
|---|---|---|---|---|---|---|---|---|---|---|
| | EN | ZH | EN | ZH | EN | ZH | EN | ZH | EN | ZH |
| *Closed-source LLMs* | | | | | | | | | | |
| OpenAI-o1-mini-2024-09-12 | **47.01** | **47.61** | **40.13** | **35.74** | **61.35** | **56.04** | **54.30** | **56.45** | **46.79** | **51.43** |
| GPT-4o-2024-08-06 | 37.68 | 36.84 | 31.03 | 24.45 | 36.71 | 42.03 | 38.71 | 39.78 | 38.21 | 40.00 |
| GPT-4o-mini-2024-07-18 | 29.19 | 27.27 | 20.69 | 20.06 | 26.09 | 22.22 | 29.84 | 28.49 | 29.11 | 27.50 |
| *Open-source Chat LLMs* | | | | | | | | | | |
| Yi-1.5-6B-Chat | 12.08 | 9.09 | 6.27 | 4.08 | 10.63 | 5.80 | 14.78 | 9.41 | 9.64 | 5.54 |
| Qwen2.5-7B-Instruct | 26.67 | 23.68 | 20.06 | 15.36 | 20.77 | 14.01 | 28.76 | 25.27 | 19.82 | 16.79 |
| LLaMA3.1-8B-Instruct | 12.68 | 8.37 | 11.29 | 4.39 | 9.66 | 5.31 | 16.13 | 8.87 | 12.14 | 8.57 |
| Ministral-8B-Instruct-2410 | 14.00 | 12.56 | 11.91 | 8.46 | 9.66 | 8.70 | 17.47 | 12.63 | 14.29 | 9.11 |
| Yi-1.5-9B-Chat | 17.58 | 12.80 | 10.66 | 8.46 | 14.01 | 10.14 | 20.16 | 15.32 | 14.64 | 10.18 |
| Mistral-Nemo-Instruct-2407 | 13.88 | 10.29 | 12.23 | 11.29 | 12.56 | 8.70 | 15.86 | 13.98 | 14.82 | 13.39 |
| DeepSeek-MOE-16B-Chat | 3.71 | 2.87 | 3.45 | 2.19 | 2.90 | 3.86 | 4.30 | 5.11 | 3.93 | 2.68 |
| DeepSeek-V2-Lite-Chat | 6.46 | 5.38 | 5.64 | 3.13 | 4.35 | 3.38 | 8.60 | 7.53 | 6.43 | 4.11 |
| Mistral-Small-Instruct-2409 | 22.61 | 19.74 | 20.69 | 15.67 | 15.94 | 16.91 | 22.85 | 23.12 | 26.43 | 27.14 |
| Yi-1.5-34B-Chat | 20.33 | 14.83 | 13.48 | 7.52 | 22.71 | 11.59 | 21.77 | 17.47 | 16.07 | 12.50 |
| LLaMA3.1-70B-Instruct | 28.23 | 25.48 | 24.45 | 19.12 | 27.05 | 21.26 | 31.72 | 28.23 | 27.14 | 26.43 |
| LLaMA3.3-70B-Instruct | 32.66 | 26.56 | 28.21 | 16.61 | 32.37 | 22.71 | 41.67 | 29.84 | 33.21 | 28.39 |
| Qwen2.5-72B-Instruct | 38.52 | 34.33 | 36.05 | 25.71 | 38.16 | 33.82 | 36.29 | 41.13 | 31.07 | 40.54 |
| Mistral-Large-Instruct-2407 | 36.72 | 34.33 | 35.74 | 28.53 | 38.65 | 37.68 | 42.74 | 41.13 | 38.04 | 41.61 |
| *Specialized Mathematical LLMs* | | | | | | | | | | |
| DeepSeek-Math-7B-Instruct | 14.47 | 12.92 | 11.60 | 9.72 | 11.11 | 9.18 | 12.90 | 13.71 | 13.39 | 13.21 |
| DeepSeek-Math-7B-RL | 15.55 | 12.68 | 13.79 | 8.46 | 10.63 | 9.18 | 18.55 | 12.37 | 14.82 | 12.50 |
| NuminaMath-7B-CoT | 14.23 | 18.42 | 13.17 | 11.91 | 8.21 | 10.14 | 14.78 | 16.13 | 14.11 | 14.82 |
| Mathstral-7B-v0.1 | 15.79 | 13.04 | 12.23 | 10.34 | 13.04 | 7.73 | 14.52 | 12.63 | 15.71 | 14.46 |
| OpenMath2-Llama-3.1-8B | 10.05 | 6.94 | 5.64 | 4.08 | 4.35 | 4.83 | 11.02 | 7.53 | 8.21 | 6.25 |
| Qwen2.5-Math-7B-Instruct | 26.08 | 25.72 | 21.00 | 19.12 | 23.19 | 16.43 | 26.61 | 20.70 | 20.54 | 17.14 |
| OpenMath2-Llama-3.1-70B | 21.05 | 20.10 | 15.99 | 16.61 | 16.43 | 14.98 | 23.39 | 20.70 | 21.07 | 17.86 |
| Qwen2.5-Math-72B-Instruct | 42.22 | 39.83 | 36.36 | 35.42 | 43.96 | 42.51 | 37.10 | 38.98 | 37.32 | 41.96 |
| *o1-like LLMs* | | | | | | | | | | |
| DeepSeek-R1-Distill-Qwen-7B | 33.13 | 23.56 | 25.08 | 17.24 | 29.95 | 28.02 | 30.38 | 18.01 | 24.82 | 17.14 |
| Skywork-o1-Open-Llama-3.1-8B | 17.46 | 13.88 | 10.03 | 5.96 | 7.25 | 5.31 | 16.40 | 9.14 | 9.82 | 6.96 |
| DeepSeek-R1-Distill-Llama-8B | 17.70 | 7.30 | 9.40 | 4.39 | 21.26 | 9.66 | 17.20 | 8.33 | 15.89 | 6.79 |
| QwQ-32B-Preview | 37.44 | 39.95 | 33.54 | 29.78 | 43.96 | 43.96 | 40.05 | 42.47 | 33.04 | 34.64 |
| DeepSeek-R1-Distill-Qwen-32B | 37.44 | 29.07 | 28.53 | 22.57 | 42.03 | 40.10 | 35.48 | 31.18 | 33.04 | 25.00 |
| DeepSeek-R1-Distill-Llama-70B | 44.26 | 33.97 | 38.24 | 30.09 | 50.24 | 37.68 | 45.70 | 40.86 | 40.36 | 31.43 |

*Table 18.* **Results across subjects of Electromagnatism on UGPhysics** (all figures are in %). Models are classified into four different categories according to their purpose and origin. The best results within each column are **bolded** and the best results of LLMs within a similar group are underlined.

| Models | Class. Elec. | | Elec.Dy. | | Geo. Optics | | Wave Optics | |
|---|---|---|---|---|---|---|---|---|
| | EN | ZH | EN | ZH | EN | ZH | EN | ZH |
| *Closed-source LLMs* | | | | | | | | |
| OpenAI-o1-mini-2024-09-12 | 45.38 | **48.21** | **41.85** | **40.76** | 27.59 | **31.03** | 45.36 | **40.40** |
| GPT-4o-2024-08-06 | 38.46 | 40.77 | 32.07 | 35.87 | 20.69 | 25.86 | 39.40 | 27.48 |
| GPT-4o-mini-2024-07-18 | 25.13 | 26.41 | 21.20 | 17.93 | 18.97 | 20.69 | 27.81 | 20.20 |
| *Open-source Chat LLMs* | | | | | | | | |
| Yi-1.5-6B-Chat | 12.56 | 8.21 | 8.70 | 8.70 | 8.62 | 12.07 | 13.91 | 5.96 |
| Qwen2.5-7B-Instruct | 26.15 | 20.00 | 17.39 | 18.48 | 20.69 | 15.52 | 26.16 | 15.56 |
| LLaMA3.1-8B-Instruct | 12.05 | 9.74 | 14.67 | 10.87 | 15.52 | 8.62 | 16.89 | 9.60 |
| Ministral-8B-Instruct-2410 | 14.87 | 6.67 | 11.96 | 11.96 | 8.62 | 12.07 | 19.87 | 9.60 |
| Yi-1.5-9B-Chat | 17.18 | 15.38 | 16.30 | 11.41 | 17.24 | 15.52 | 13.58 | 11.26 |
| Mistral-Nemo-Instruct-2407 | 15.38 | 12.82 | 16.85 | 11.41 | 13.79 | 10.34 | 12.91 | 11.59 |
| DeepSeek-MOE-16B-Chat | 3.85 | 3.85 | 4.89 | 4.35 | 5.17 | 1.72 | 6.29 | 6.95 |
| DeepSeek-V2-Lite-Chat | 7.69 | 4.87 | 6.52 | 7.07 | 6.90 | 3.45 | 5.63 | 6.62 |
| Mistral-Small-Instruct-2409 | 25.13 | 19.74 | 17.93 | 20.65 | 17.24 | 15.52 | 23.51 | 15.89 |
| Yi-1.5-34B-Chat | 20.00 | 12.05 | 14.13 | 14.13 | 15.52 | 15.52 | 20.20 | 12.25 |
| LLaMA3.1-70B-Instruct | 28.46 | 24.62 | 26.09 | 23.37 | 10.34 | 17.24 | 28.81 | 21.19 |
| LLaMA3.3-70B-Instruct | 33.85 | 29.49 | 30.98 | 23.91 | 20.69 | 17.24 | 36.42 | 20.86 |
| Qwen2.5-72B-Instruct | 35.64 | 37.18 | 31.52 | 33.70 | 18.97 | 27.59 | 33.77 | 31.46 |
| Mistral-Large-Instruct-2407 | 43.33 | 40.00 | 33.15 | 34.78 | 24.14 | 29.31 | 37.75 | 29.47 |
| *Specialized Mathematical LLMs* | | | | | | | | |
| DeepSeek-Math-7B-Instruct | 17.69 | 14.87 | 13.04 | 11.41 | 17.24 | 10.34 | 16.89 | 12.58 |
| DeepSeek-Math-7B-RL | 17.44 | 14.62 | 17.93 | 10.87 | 13.79 | 12.07 | 10.93 | 11.59 |
| NuminaMath-7B-CoT | 15.90 | 15.13 | 15.22 | 15.22 | 10.34 | 22.41 | 12.91 | 14.90 |
| Mathstral-7B-v0.1 | 16.41 | 13.08 | 18.48 | 19.02 | 15.52 | 15.52 | 19.54 | 14.90 |
| OpenMath2-Llama-3.1-8B | 10.00 | 6.15 | 8.70 | 7.61 | 13.79 | 8.62 | 10.60 | 8.61 |
| Qwen2.5-Math-7B-Instruct | 24.62 | 22.56 | 19.57 | 13.59 | 15.52 | 13.79 | 22.52 | 15.89 |
| OpenMath2-Llama-3.1-70B | 21.54 | 21.28 | 23.91 | 15.76 | 17.24 | 25.86 | 22.85 | 15.56 |
| Qwen2.5-Math-72B-Instruct | 40.26 | 40.77 | 28.26 | 38.59 | **29.31** | 22.41 | 36.09 | 38.41 |
| *o1-like LLMs* | | | | | | | | |
| DeepSeek-R1-Distill-Qwen-7B | 32.05 | 20.51 | 18.48 | 13.59 | 15.52 | 10.34 | 29.14 | 17.22 |
| Skywork-o1-Open-Llama-3.1-8B | 11.54 | 8.97 | 14.13 | 9.78 | 20.69 | 6.90 | 17.22 | 7.95 |
| DeepSeek-R1-Distill-Llama-8B | 18.46 | 6.92 | 12.50 | 7.07 | 13.79 | 6.90 | 18.87 | 5.63 |
| QwQ-32B-Preview | 35.38 | 35.13 | 31.52 | 36.96 | 20.69 | 22.41 | 41.39 | 25.50 |
| DeepSeek-R1-Distill-Qwen-32B | 37.44 | 27.44 | 22.28 | 21.74 | 15.52 | 17.24 | 38.41 | 20.20 |
| DeepSeek-R1-Distill-Llama-70B | **46.41** | 31.54 | 34.78 | 28.80 | 25.86 | 29.31 | **46.69** | 25.17 |

*Table 19.* **Results across subjects of Modern Physics on UGPhysics** (all figures are in %). Models are classified into four different categories according to their purpose and origin. The best results within each column are **bolded** and the best results of LLMs within a similar group are underlined.

| Models | Quan. Mech. | | Atomic Phy. | | S.-S. Phy. | | Semi. Phy. | |
|---|---|---|---|---|---|---|---|---|
| | EN | ZH | EN | ZH | EN | ZH | EN | ZH |
| *Closed-source LLMs* | | | | | | | | |
| OpenAI-o1-mini-2024-09-12 | **49.95** | **50.54** | **58.58** | **54.86** | **43.60** | **45.93** | **63.98** | **60.75** |
| GPT-4o-2024-08-06 | 41.71 | 39.06 | 44.48 | 39.23 | 36.05 | 41.28 | 46.77 | 58.94 |
| GPT-4o-mini-2024-07-18 | 27.28 | 28.66 | 33.99 | 28.63 | 27.91 | 27.33 | 36.02 | 37.63 |
| *Open-source Chat LLMs* | | | | | | | | |
| Yi-1.5-6B-Chat | 11.48 | 7.56 | 15.19 | 9.84 | 9.30 | 8.72 | 22.04 | 16.67 |
| Qwen2.5-7B-Instruct | 23.06 | 20.22 | 27.10 | 21.75 | 22.09 | 21.51 | 34.95 | 30.65 |
| LLaMA3.1-8B-Instruct | 13.05 | 10.79 | 19.89 | 14.75 | 10.47 | 11.05 | 27.96 | 18.28 |
| Ministral-8B-Instruct-2410 | 15.90 | 10.21 | 21.86 | 12.68 | 15.70 | 11.63 | 27.42 | 20.97 |
| Yi-1.5-9B-Chat | 16.58 | 14.82 | 22.84 | 14.43 | 14.53 | 14.53 | 29.03 | 24.19 |
| Mistral-Nemo-Instruct-2407 | 16.68 | 14.33 | 20.44 | 17.38 | 15.12 | 13.37 | 20.97 | 24.19 |
| DeepSeek-MOE-16B-Chat | 5.00 | 4.22 | 9.73 | 5.03 | 6.40 | 1.74 | 6.99 | 5.91 |
| DeepSeek-V2-Lite-Chat | 6.67 | 6.48 | 13.01 | 9.07 | 2.91 | 4.07 | 13.44 | 9.68 |
| Mistral-Small-Instruct-2409 | 27.67 | 23.06 | 31.91 | 22.40 | 25.58 | 21.51 | 37.10 | 29.57 |
| Yi-1.5-34B-Chat | 18.84 | 18.06 | 26.56 | 17.27 | 22.09 | 11.05 | 31.18 | 25.81 |
| LLaMA3.1-70B-Instruct | 31.11 | 27.38 | 33.22 | 26.01 | 29.65 | 24.42 | 45.16 | 34.95 |
| LLaMA3.3-70B-Instruct | 34.54 | 26.30 | 43.83 | 25.36 | 36.63 | 27.33 | 44.09 | 36.56 |
| Qwen2.5-72B-Instruct | 34.25 | 37.98 | 40.00 | 37.81 | 33.14 | 34.30 | 42.47 | 45.16 |
| Mistral-Large-Instruct-2407 | 42.39 | 41.71 | 47.10 | 38.69 | 34.30 | 36.05 | 48.92 | 45.16 |
| *Specialized Mathematical LLMs* | | | | | | | | |
| DeepSeek-Math-7B-Instruct | 15.51 | 13.15 | 23.17 | 16.39 | 15.70 | 17.44 | 21.51 | 23.66 |
| DeepSeek-Math-7B-RL | 16.98 | 12.76 | 20.44 | 17.81 | 15.12 | 12.21 | 20.97 | 20.97 |
| NuminaMath-7B-CoT | 17.27 | 16.19 | 18.91 | 20.22 | 12.21 | 14.53 | 23.66 | 26.34 |
| Mathstral-7B-v0.1 | 16.88 | 16.00 | 20.77 | 17.81 | 23.26 | 12.21 | 29.57 | 23.66 |
| OpenMath2-Llama-3.1-8B | 9.91 | 8.34 | 13.33 | 10.16 | 11.05 | 8.14 | 17.20 | 14.52 |
| Qwen2.5-Math-7B-Instruct | 24.44 | 24.14 | 28.74 | 18.58 | 19.77 | 16.28 | 33.33 | 24.19 |
| OpenMath2-Llama-3.1-70B | 21.10 | 18.25 | 28.52 | 22.62 | 28.49 | 21.51 | 31.18 | 33.87 |
| Qwen2.5-Math-72B-Instruct | 39.35 | 41.22 | 43.61 | 37.60 | 31.98 | 34.30 | 47.85 | 43.55 |
| *o1-like LLMs* | | | | | | | | |
| DeepSeek-R1-Distill-Qwen-7B | 27.67 | 21.00 | 32.35 | 20.33 | 19.77 | 12.21 | 38.17 | 28.49 |
| Skywork-o1-Open-Llama-3.1-8B | 12.95 | 9.52 | 17.70 | 10.38 | 11.63 | 7.56 | 20.43 | 14.52 |
| DeepSeek-R1-Distill-Llama-8B | 16.39 | 9.13 | 25.57 | 9.07 | 11.05 | 6.98 | 31.72 | 10.75 |
| QwQ-32B-Preview | 33.56 | 37.88 | 45.36 | 40.87 | 27.33 | 28.49 | 43.55 | 44.62 |
| DeepSeek-R1-Distill-Qwen-32B | 32.78 | 26.10 | 44.70 | 31.48 | 26.16 | 21.51 | 45.70 | 37.10 |
| DeepSeek-R1-Distill-Llama-70B | 46.71 | 35.82 | 53.66 | 35.63 | 40.70 | 36.05 | 57.53 | 48.39 |

*Table 20.* **Results across different skill sets on UGPhysics** (all figures are in %). Models are classified into four different categories according to their purpose and origin. The best results within each column are **bolded** and the best results of LLMs within a similar group are underlined.

| Models | Know. Recall | | Laws App. | | Math Deri. | | Prac. App. | | Others | |
|---|---|---|---|---|---|---|---|---|---|---|
| | EN | ZH | EN | ZH | EN | ZH | EN | ZH | EN | ZH |
| *Closed-source LLMs* | | | | | | | | | | |
| OpenAI-o1-mini-2024-09-12 | **69.18** | **63.87** | **53.60** | **51.46** | **42.16** | **45.05** | **47.92** | **45.75** | **50.49** | **50.97** |
| GPT-4o-2024-08-06 | 63.36 | 55.25 | 42.46 | 38.94 | 31.40 | 33.71 | 34.54 | 32.72 | 38.83 | 40.69 |
| GPT-4o-mini-2024-07-18 | 53.77 | 48.29 | 30.81 | 28.00 | 20.72 | 22.48 | 24.59 | 17.00 | 29.61 | 26.70 |
| *Open-source Chat LLMs* | | | | | | | | | | |
| Yi-1.5-6B-Chat | 26.37 | 21.23 | 14.46 | 7.97 | 6.71 | 5.81 | 10.67 | 4.34 | 15.53 | 9.71 |
| Qwen2.5-7B-Instruct | 48.29 | 40.24 | 26.78 | 21.72 | 17.48 | 15.68 | 19.71 | 13.92 | 27.18 | 22.33 |
| LLaMA3.1-8B-Instruct | 34.93 | 25.51 | 15.99 | 10.83 | 8.02 | 6.44 | 14.10 | 6.51 | 17.48 | 12.62 |
| Ministral-8B-Instruct-2410 | 37.67 | 26.37 | 16.91 | 10.42 | 10.41 | 8.65 | 14.83 | 5.61 | 19.90 | 14.56 |
| Yi-1.5-9B-Chat | 36.13 | 29.62 | 20.49 | 13.64 | 10.95 | 9.95 | 14.29 | 9.40 | 18.45 | 16.02 |
| Mistral-Nemo-Instruct-2407 | 35.62 | 32.71 | 15.94 | 12.62 | 11.58 | 10.50 | 12.84 | 8.50 | 16.99 | 16.50 |
| DeepSeek-MOE-16B-Chat | 13.18 | 9.42 | 5.52 | 4.45 | 3.06 | 2.43 | 4.88 | 2.71 | 7.77 | 4.85 |
| DeepSeek-V2-Lite-Chat | 22.26 | 17.98 | 8.94 | 6.34 | 3.78 | 3.74 | 4.16 | 3.25 | 8.25 | 5.34 |
| Mistral-Small-Instruct-2409 | 50.17 | 41.78 | 25.45 | 20.75 | 20.54 | 18.74 | 19.53 | 12.48 | 31.55 | 27.67 |
| Yi-1.5-34B-Chat | 42.81 | 33.56 | 23.91 | 15.59 | 12.43 | 10.81 | 17.36 | 11.93 | 22.33 | 13.59 |
| LLaMA3.1-70B-Instruct | 52.23 | 42.47 | 30.76 | 24.83 | 23.87 | 23.06 | 27.12 | 18.81 | 29.61 | 28.16 |
| LLaMA3.3-70B-Instruct | 59.25 | 40.92 | 37.71 | 24.48 | 28.47 | 24.37 | 34.54 | 20.25 | 35.44 | 33.01 |
| Qwen2.5-72B-Instruct | 56.68 | 56.34 | 38.02 | 37.61 | 29.01 | 32.39 | 33.27 | 26.76 | 40.29 | 39.32 |
| Mistral-Large-Instruct-2407 | 64.73 | 58.73 | 42.51 | 38.17 | 33.56 | 34.86 | 37.43 | 27.12 | 39.81 | 38.35 |
| *Specialized Mathematical LLMs* | | | | | | | | | | |
| DeepSeek-Math-7B-Instruct | 37.33 | 32.36 | 16.96 | 14.05 | 10.50 | 10.27 | 13.02 | 6.69 | 19.42 | 16.99 |
| DeepSeek-Math-7B-RL | 38.36 | 30.14 | 16.35 | 12.62 | 12.39 | 10.27 | 9.58 | 10.67 | 20.87 | 14.56 |
| NuminaMath-7B-CoT | 32.36 | 33.90 | 14.51 | 18.14 | 12.57 | 11.85 | 12.12 | 11.03 | 20.39 | 23.30 |
| Mathstral-7B-v0.1 | 41.10 | 28.94 | 16.66 | 14.41 | 12.70 | 12.43 | 13.56 | 9.76 | 19.42 | 17.48 |
| OpenMath2-Llama-3.1-8B | 21.92 | 18.66 | 11.24 | 8.07 | 6.71 | 4.95 | 7.78 | 7.23 | 13.11 | 7.28 |
| Qwen2.5-Math-7B-Instruct | 45.21 | 35.10 | 26.26 | 20.34 | 18.87 | 18.78 | 20.61 | 11.39 | 25.73 | 28.16 |
| OpenMath2-Llama-3.1-70B | 47.60 | 37.67 | 22.79 | 21.67 | 16.67 | 14.10 | 19.89 | 17.54 | 25.24 | 20.39 |
| Qwen2.5-Math-72B-Instruct | 61.30 | 56.51 | 41.44 | 38.83 | 33.69 | 38.15 | 33.82 | 28.57 | 39.81 | 39.81 |
| *o1-like LLMs* | | | | | | | | | | |
| DeepSeek-R1-Distill-Qwen-7B | 47.26 | 34.59 | 32.40 | 20.95 | 22.03 | 16.53 | 26.40 | 15.73 | 31.55 | 21.36 |
| Skywork-o1-Open-Llama-3.1-8B | 31.16 | 21.58 | 16.04 | 9.04 | 9.01 | 7.61 | 12.12 | 6.51 | 16.02 | 11.65 |
| DeepSeek-R1-Distill-Llama-8B | 39.55 | 19.86 | 21.15 | 8.28 | 10.68 | 4.91 | 16.46 | 4.52 | 19.90 | 10.19 |
| QwQ-32B-Preview | 58.05 | 57.19 | 41.29 | 39.04 | 28.15 | 31.85 | 36.53 | 31.46 | 43.20 | 38.83 |
| DeepSeek-R1-Distill-Qwen-32B | 56.85 | 45.89 | 39.29 | 30.10 | 27.61 | 22.07 | 37.25 | 22.24 | 35.44 | 30.10 |
| DeepSeek-R1-Distill-Llama-70B | 64.38 | 46.06 | 49.41 | 36.02 | 38.56 | 31.35 | 43.04 | 28.39 | 48.54 | 34.47 |

**Problem**: A crystal has $N$ atoms and possesses Frenkel defects. The numbers of normal positions and interstitial positions are $N$ and $N'$ respectively. The energy when an atom is in normal and interstitial positions is $\varepsilon_1$ and $\varepsilon_2$, with $\varepsilon_2 - \varepsilon_1 = w > 0$. Derive the function relationship of the most probable number of interstitial atoms $n$ as a function of temperature $T$ (assuming $n, N, N' \gg 1$).

**o1-mini Completion**: To determine the most probable number of interstitial atoms $n$ as a function of temperature $T$, we consider the statistical mechanics of Frenkel defects in a crystal.

1. **Multiplicity ($\Omega$)**: The number of ways to arrange $n$ interstitial atoms among $N$ normal sites and $N'$ interstitial sites is given by:

$$\Omega = \binom{N}{n}\binom{N'}{n}$$

[should be $\Omega = \binom{N}{N-n}\binom{N'}{n}$]

Taking the natural logarithm and using Stirling's approximation for large $n, N, N'$:

$$\ln \Omega \approx n \ln\left(\frac{N}{n}\right) + n \ln\left(\frac{N'}{n}\right)$$

2. **Energy ($U$)**: The total energy associated with $n$ interstitial atoms is:

$$U = N\varepsilon_1 + n(\varepsilon_2 - \varepsilon_1) = N\varepsilon_1 + nw$$

Here, $w = \varepsilon_2 - \varepsilon_1$.

3. **Free Energy Maximization**: To find the most probable $n$, we maximize the entropy subject to the energy constraint:

$$\frac{d}{dn}\left(\ln \Omega - \frac{U}{kT}\right) = 0$$

[should be $\delta \ln W - \alpha \sum_{l=1}^{2} \delta N_l - \beta \sum_{l=1}^{2} \varepsilon_l \delta N_l = 0$], applying Lagrange multiplier method, where $N_1 = N - n$, $N_2 = n$]

This leads to:

$$\ln\left(\frac{NN'}{n^2}\right) - 2 - \frac{w}{kT} = 0$$

Solving for $n$:

$$\ln n = \frac{1}{2}\left(\ln(NN') - 2 - \frac{w}{kT}\right)$$

Exponentiating both sides gives:

$$n = \sqrt{NN'}\, e^{-w/(2kT)}$$

So the final answer is $\boxed{\sqrt{NN'}\, e^{-w/(2kT)}}$.

**Ground-Truth Answer**:

$$\exp\left(-\frac{w}{kT}\right)$$

**Model Output**:

$$\sqrt{NN'}e^{-\frac{w}{2kT}}$$

*Table 21.* An example of Statistical Mechanics with flawed reasoning error.

**Problem**: An electromagnetic eddy current brake consists of a disc with electrical conductivity $\sigma$ and thickness $t$, which rotates around an axis through its center at an angular velocity $\omega$. There is a magnetic field $\boldsymbol{B}$ with a coverage area of $a^2$ perpendicular to the disc. The area $a^2$ is located at a distance $r$ from the axis. Determine the magnitude of the eddy current $I$ generated under this configuration.

**o1-mini Completion**: To determine the magnitude of the eddy current $I$ generated in the electromagnetic eddy current brake, we follow these steps:

1. **Induced Electric Field:** The disc rotates with an angular velocity $\omega$ in a uniform magnetic field $\boldsymbol{B}$ perpendicular to its surface. The tangential velocity at a distance $r$ from the center is $\boldsymbol{v} = \omega r \hat{\theta}$. The induced electric field $\boldsymbol{E}$ due to this motion is given by:

$$\boldsymbol{E} = \boldsymbol{v} \times \boldsymbol{B} = \omega r B \hat{r}$$

2. **Current Density:** The current density $\boldsymbol{J}$ is related to the electric field by Ohm's law:

$$\boldsymbol{J} = \sigma \boldsymbol{E} = \sigma \omega B r \hat{r}$$

3. **Total Eddy Current:** To find the total current $I$, we integrate the current density over the area where the magnetic field is present. Considering the thickness $t$ of the disc and integrating from the center ($r = 0$) to the extent of the magnetic field coverage ($r = a$):

$$I = \int_0^a \boldsymbol{J} \cdot d\boldsymbol{A} = \int_0^a \sigma \omega B r \cdot (2\pi r t) \, dr$$

[should be $I = \frac{\varepsilon}{R} = \frac{r\omega B a}{\frac{a}{\sigma a t}} = r\omega B a \sigma t$]

Simplifying the integral:

$$I = 2\pi\sigma\omega B t \int_0^a r^2 \, dr = 2\pi\sigma\omega B t \left[\frac{a^3}{3}\right] = \frac{2\pi}{3}\sigma\omega B t a^3$$

However, considering the standard configuration and simplifying constants for the practical scenario, the magnitude of the eddy current can be expressed as:

$$I = \pi\sigma\omega B t a^2$$

So the final answer is $\boxed{\pi\sigma\omega B t a^2}$ (A).

**Ground-Truth Answer**: $r\omega B a \sigma t$

**Model Output**: $\pi\sigma\omega B t a^2$

*Table 22.* An example of Classical Electromagnetism with wrong application error.

**Problem**: Calculate the effective mass of a hole $m_h^*$. It is known that the static dielectric constant of InP is 9.6, $E_g = 1.400\,\mathrm{eV}$, $E_C = 1.386\,\mathrm{eV}$, and holes in InP are much lighter than electrons ($m_h^* \ll m_e^*$).

**o1-mini Completion**: To calculate the effective mass of a hole $m_h^*$ in InP, we can use the effective Rydberg formula adapted for semiconductors. The effective Rydberg energy $R^*$ is given by:

$$R^* = \frac{\mu e^4}{2(4\pi\epsilon_0\epsilon)^2\hbar^2}$$

where: - $\mu$ is the reduced mass, which for $m_h^* \ll m_e^*$ simplifies to $\mu \approx m_h^*$. - $e$ is the elementary charge. - $\epsilon_0$ is the vacuum permittivity. - $\epsilon$ is the static dielectric constant. - $\hbar$ is the reduced Planck's constant.
Assuming the exciton binding energy $E_b$ in InP is approximately $0.01\,\mathrm{eV}$, we can solve for the reduced mass $\mu$:

$$\mu = \frac{2R^*(4\pi\epsilon_0\epsilon)^2\hbar^2}{e^4}$$

Given that $R^* \approx E_b = 0.01\,\mathrm{eV}$, and substituting the known values:

$$\color{red}\mu \approx m_h^* = \frac{0.01 \times 9.6^2}{13.6} \approx 0.07\,m_0$$

So the final answer is $\boxed{0.07}$.
Correct knowledge should look like this:
For hydrogen-like cases, the distance of these absorption lines from the fundamental absorption threshold is:

$$E_n = E_g - \frac{m_r^* e^4}{8\varepsilon^2\varepsilon_0^2 h^2 n^2}$$

where $m_r^*$ is the reduced mass of the electron-hole pair. Since $m_h^* \ll m_e^*$, it follows that:

$$m_r^* = \frac{m_e^* m_h^*}{m_e^* + m_h^*} \approx m_h^*$$

For the ground state, $n = 1$, and $E_1 = E_C = 1.386\,\mathrm{eV}$, thus:

$$
\begin{aligned}
m_h^* &= \frac{8\varepsilon^2\varepsilon_0^2 h^2 (E_g - E_C)}{e^4} \\
&= \frac{8 \times 9.6^2 \times (8.854 \times 10^{-12})^2 \times (6.626 \times 10^{-34})^2 \times 0.014 \times 1.6 \times 10^{-19}}{(1.6 \times 10^{-19})^4} \\
&= 8.67 \times 10^{-36}\,\mathrm{kg} = \frac{8.67 \times 10^{-36}}{9.1 \times 10^{-31}} \approx 9.52 \times 10^{-2} m_0
\end{aligned}
$$

**Ground-Truth Answer**: $9.52 \times 10^{-2}$

**Model Output**: $0.07$

*Table 23.* An example of Semiconductor Physics with knowledge deficiency error.

