# OpenReview forum: "UGPhysics: A Comprehensive Benchmark for Undergraduate Physics Reasoning with Large Language Models"
_ICML.cc/2025/Conference — ICML 2025 poster_

### Official Review · Reviewer_wqeU · 2025-03-03

**Overall Recommendation:** 3

**Summary:**

The paper proposes a new benchmark for reasoning in physics by LLMs, they used 31 LLMs to evaluate the performance on the proposed benchmark, then they have introduced a new method MARJ to evaluate the outputs on these 31 LLM on the benchmark. Overall they show that OpenAI-01-mini gives the best performance on this benchmark.

**Claims And Evidence:**

**Claim:** Introduction of UGPhysics, benchmark to evaluate the Physics reasoning with LLMs,

**Evidence:** Though they do introduce a benchmark, it’s not totally about reasoning, it’s also about problem solving abilities.

**Claim:** Introduction of MARJ for evaluating the outputs given by different LLMs on the benchmark.

**Evidence:** They certainly do introduce MARJ, and give details of the method. In section 3.3.

**Claim:** Open Ai-o1 Achieves the best score on this benchmark:

**Evidence:** Table 5 shows the results of all the models and OpenAI-01 scores the highest.

**Essential References Not Discussed:**

The related work is well discussed, they mention the existing work related to Physics, existing benchmarks related to Physics and also reasoning.

**Experimental Designs Or Analyses:**

The authors have considered 31 leading models, but the choice of models is not clear, as in the closed source LLM only one family of models are considered, there are other families of closed source models which I feel should have been considered for better evaluation. Also models of Phi family which are specifically trained on text book data are not considered. The selection of models according to the task should be considered even if it is a lesser number of models.

**Methods And Evaluation Criteria:**

The evaluation method used doesn’t really support this work, the human evaluation of the proposed method was only checked with 100 examples, which I feel is a very small sample, apart from this the model used for evaluation is Open AI-4o which in turn is the same family of models that shows the best performance.

**Other Comments Or Suggestions:**

N/A

**Other Strengths And Weaknesses:**

**Strengths**:

New benchmark for evaluating how well LLMs can work on Physics problems

Evaluating the performance of 31 different models on the proposed benchmark

**Weaknesses**:

The Evaluation method is not robust enough, as mentioned above in Methods And Evaluation Criteria*

The paper writing has a good amount of redundancy, specifically the section 3.1, which was mentioned multiple times and also about the evaluation method.

Until line 81 there is not mention of language of the data and then there is a mention of translating into English

In line 87 they say rigorous data leakage, but the methods to check data leakage were not that rigorous. More details on checking the data leakage should be provided, like the number of times each problem was run and then checked if in every case the model didn’t give the output as in the question then you could say there is no data leakage.

In line 176, there is no reference to the books used for data creation.

In the appendix, more examples and one whole example of whole work from textbook to translating to getting output from LLM and then evaluation of it with ground truth would be better.

**Questions For Authors:**

N/A

**Relation To Broader Scientific Literature:**

If mentioned weaknesses are corrected, the benchmark can be of use for fine tuning and working with LLMs for science.

**Theoretical Claims:**

There aren’t any theoretical claims

---

> ### Author Rebuttal · Authors · 2025-03-29
>
> Dear wqeU,
>
> Thank you for your time and effort to review our work! We will reply to your questions one by one as follows:
>
> >  the human evaluation of the proposed method was only checked with 100 examples, which I feel is a very small sample, apart from this the model used for evaluation is Open AI-4o which in turn is the same family of models that shows the best performance.
>
> Previous studies [1, 2, 3] have shown that using similar or even smaller sample sizes (100 [1], 80 [2], and 50 [3]) is sufficient for human evaluation tasks, even for more subjective tasks such as text summarization [3]. The second-best LLMs  (QWQ and DS-Distill) are not within the same family of GPT-4o. Additionally, OpenAI o1-mini is a Long CoT LLM, which is quite different from GPT-4o. [1] has also shown that LLMs-as-judge is a valid approach even if the LLMs are within the same family.
>
> > The authors have considered 31 leading models, but the choice of models is not clear, as in the closed source LLM only one family of models are considered, there are other families of closed source models which I feel should have been considered for better evaluation. Also models of Phi family which are specifically trained on text book data are not considered. The selection of models according to the task should be considered even if it is a lesser number of models.
>
> Thank you for your suggestion to include Phi. We will **add the results of Phi-4 as follows**:
>
> | Mec. and Ther. (EN) | Mec. and Ther. (ZH) | Elec. (EN) | Elec. (ZH) | Modern Physics (EN) | Modern Physics (ZH) | Overall (EN) | Overall (ZH) | Average  |
> |---------------------|---------------------|------------|------------|----------------------|---------------------|--------------|--------------|----------|
> | 0.3413              | 0.3248              | 0.3651     | 0.2987     | 0.4045               | 0.3586              | 0.3716       | 0.3344       | 0.3530   |
>
>
> From the results, Phi-4 is a very strong fast-thinking LLM.
>
> We have listed all the details of the chosen LLMs in Appendix B.1. We have covered OpenAI, Qwen, Llama, DeepSeek, Mistral, Skywork, Yi, Numina, and OpenMath2, which we believe are very diverse. It is quite expensive to cover more closed-source LLMs, especially Claude-series, and we could not afford to do so for more closed-source LLMs.
>
>
> > The paper writing has a good amount of redundancy, specifically the section 3.1, which was mentioned multiple times and also about the evaluation method.
>
> Thank you for your suggestion, we will **change the name of Section 3.1** to “UGPhysics and MARJ Overview”.
>
> > In line 87 they say rigorous data leakage, but the methods to check data leakage were not that rigorous. More details on checking the data leakage should be provided, like the number of times each problem was run and then checked if in every case the model didn’t give the output as in the question then you could say there is no data leakage.
>
> Thank you for pointing this out. We will **change our wording about "rigorous"**. This data leakage detection [4] is widely adopted [1, 5], and is believed to be in some sense useful.  All our settings align with [5]. Although this method is not perfect, we believe that **conducting such detection is a merit rather than a shortage**.
>
> > In line 176, there is no reference to the books used for data creation.
>
> Thank you for your question! There is a risk to reveal the institute of several authors and we give the links to these books at a later stage (if possible).
>
> > In the appendix, more examples and one whole example of whole work from textbook to translating to getting output from LLM and then evaluation of it with ground truth would be better.
>
> Thank you for your comments. We will consider adding more examples to the appendix and add a whole example as well.
>
>
> Thank you again for your effort and suggestions. We hope our rebuttal has addressed your concerns. Feel free to discuss if you have any further questions or comments.
>
> Sincerely,
>
> Authors
>
> [1] Gao et al., 2024; Omni-Math: A Universal Olympiad-Level Mathematics Benchmark for Large Language Models.
>
> [2] Shaib et al., 2024; How Much Annotation is Needed to Compare Summarization Models.
>
> [3] Zheng et al., 2023; Judging LLM-as-a-Judge with MT-Bench and Chatbot Arena.
>
> [4] Xu et al., 2024; Benchmarking Benchmark Leakage in Large Language Models.
>
> [5] Huang et al., 2024; OlympicArena: Benchmarking Multi-Discipline Cognitive Reasoning for Superintelligent AI.

---

> > ### Comment · Reviewer_wqeU · 2025-04-03
> >
> > Thank you for the considerations, these additions would definitely help strengthen the paper, after going through the rebuttal, I would like to increase my score.

---

### Official Review · Reviewer_SPQT · 2025-03-03

**Overall Recommendation:** 4

**Summary:**

The paper introduces a comprehensive bilingual benchmark UGPhysics for evaluating undergraduate physics reasoning, featuring 5520 questions across 13 subjects. The benchmark also comes with a proposed evaluation pipeline that combines rule-based and model-based methods for improved accuracy. Notably, the study finds that even top-performing LLMs achieve less than 50% accuracy on this proposed benchmark, highlighting a critical need for improvement in LLM capabilities for physics reasoning.


## update after rebuttal
I have read the author response. I'll keep my score.

**Claims And Evidence:**

The proposed benchmark is well-curated and manually reviewed.

**Essential References Not Discussed:**

The paper covers related work well.

**Experimental Designs Or Analyses:**

The experiments that examine 31 leading LLMs' performance on UGPhysics are well executed.

**Methods And Evaluation Criteria:**

The proposed evaluation pipeline MARJ makes sense to me. However, one concern I have is that in 5.2 Reliability of Evaluation, only 100 random test examples are being examined and it's not clear what are the answer types of those questions. Do they cover all the answer types, or only some of the seven answer types?

**Other Comments Or Suggestions:**

It would be nice to also have a table that lists the information of how many questions are there for each answer type.

**Other Strengths And Weaknesses:**

Please see other sections.

**Questions For Authors:**

- In line 372-382: "LLMs show varying performance across different subjects, although the disparity is relatively small..." I don't know where the numbers mentioned are coming from (e.g., 27.0%, 22.9%, 13.8% etc), I don't see those numbers from Figure 2 (a), could you clarify this?
- For o1 like models, I would like to know how many percentages of the inference generations didn't end due to length limit (i.e., 8192 as mentioned in the Appendix).

**Relation To Broader Scientific Literature:**

While mathematical reasoning has numerous benchmarks, AI for physics remains underexplored, lacking challenging evaluations and diverse question types. This paper addresses this gap by proposing a comprehensive physics benchmark that surpasses previous ones in size, difficulty, and subject coverage.

**Theoretical Claims:**

The paper does not have theoretical claims.

---

> ### Author Rebuttal · Authors · 2025-03-28
>
> Dear Reviewer SPQT,
>
> Thank you for your valuable suggestions! We will reply to your questions one by one as follows:
>
> > However, one concern I have is that in 5.2 Reliability of Evaluation, only 100 random test examples are being examined and it's not clear what are the answer types of those questions. Do they cover all the answer types, or only some of the seven answer types?
>
> Thank you for your question! Previous studies [1, 2, 3] have shown that using similar or even smaller sample sizes (100 [1], 80 [2], and 50 [3]) is sufficient for human evaluation tasks, even for more subjective tasks such as text summarization [3].
> Regarding the types of answers, after examining 100 randomly selected test examples, we observed that all seven answer types are represented, except for True/False (TF). We believe it is OK because the evaluation of the TF question is relatively straightforward.
>
> > In line 372-382: "LLMs show varying performance across different subjects, although the disparity is relatively small..." I don't know where the numbers mentioned are coming from (e.g., 27.0%, 22.9%, 13.8% etc), I don't see those numbers from Figure 2 (a), could you clarify this?
>
> Thank you for pointing this out! We apologize that we mistakenly put the wrong numbers after the update of Figure 2(a). We will **correct these numbers accordingly**:
> "As shown in Figure 2a, the average overall accuracy of eight strong LLMs reveals that they perform particularly well in Semiconductor Physics (31.0%) and Atomic Physics (26.7%).
> In contrast, their performance is slightly lower in Theoretical Mechanics (16.5%). Additionally, LLMs show minor performance variation across six out of 13 subjects, with accuracies hovering around 20%”
>
> > For o1 like models, I would like to know how many percentages of the inference generations didn't end due to length limit (i.e., 8192 as mentioned in the Appendix).
>
> Thank you for your insightful question! In our experiments, we found that most cases of o1-mini generation will end within 8192 tokens. From the analysis in Section 5.3, the error incurred by the length limit is around 5% in all failure cases (approximately 2.5% = 5% * 50% in total). After checking the other open-source o1-like LLMs, we find the percentage is much higher than o1-mini.
>
> We will **add the following table of this percentage as follows** (in %):
>
> | Models                        | 8192  | 16382  |
> |-------------------------------|-------|--------|
> | DeepSeek-R1-Distill-Qwen-32B  | 38.55 | 34.47  |
> | DeepSeek-R1-Distill-Qwen-7B   | 44.40 | 38.90  |
> | DeepSeek-R1-Distill-Llama-70B | 19.16 | 12.25  |
> | DeepSeek-R1-Distill-Llama-8B  | 52.37 | 43.80  |
> | o1-mini-2024-09-12            | 2.01  |     -   |
> | QwQ-32B-Preview               | 19.01 | 8.54   |
>
> We believe this is also a gap between open-source o1-like LLMs and OpenAI o1 series. We will **add a paragraph in Section 5.1 to discuss this as follows**:
>
> "Open-source o1-like LLMs typically consume more tokens compared to OpenAI's o1-mini when solving problems in UGPhysics. When the maximum length of generation is set to 8192 tokens, only around 2% of OpenAI o1-mini’s generations exceed this length limit. In contrast, a significantly higher proportion of inference generations for other open-source o1-like LLMs fail to terminate within the specified limit, as shown in the previous table. To assess whether increasing the maximum generation length improves the performance of these o1-like LLMs, we conducted additional experiments by extending the token limit to 16384. The results, presented in the following table, demonstrate that doubling the maximal generation tokens only slightly improves the performance of o1-like LLMs. Additionally, we report the proportion of cases where the generation did not terminate due to the extended length limit of 16384 tokens. These findings suggest addressing the redundancy in token consumption of o1-like LLMs [4] during reasoning remains an important direction for further research."
>
> | Models /Acc (in %)                       | 8192  | 16384 |
> |-------------------------------|-------|-------|
> | DeepSeek-R1-Distill-Qwen-7B   | 24.64 | 24.86 |
> | DeepSeek-R1-Distill-Llama-8B  | 13.11 | 14.51 |
> | QwQ-32B-Preview               | 37.34 | 38.90 |
> | DeepSeek-R1-Distill-Qwen-32B  | 31.93 | 32.21 |
> | DeepSeek-R1-Distill-Llama-70B | 40.17 | 41.77 |
>
>
>
> Thank you once again for your insightful comments to improve the quality of our work. Feel free to discuss if you have any further questions or comments.
>
> Sincerely,
>
> Authors
>
> [1] Gao et al., 2024; Omni-Math: A Universal Olympiad-Level Mathematics Benchmark for Large Language Models.
>
> [2] Shaib et al., 2024; How Much Annotation is Needed to Compare Summarization Models.
>
> [3] Zheng et al., 2023; Judging LLM-as-a-Judge with MT-Bench and Chatbot Arena.
>
> [4] Chen et al., 2024; Do Not Think That Much for 2+3=? On the Overthinking of o1-like LLMs.

---

> > ### Comment · Reviewer_SPQT · 2025-04-03
> >
> > Thanks for your response.

---

### Official Review · Reviewer_3YX8 · 2025-03-13

**Overall Recommendation:** 3

**Summary:**

This paper proposes a new benchmark that targets underground-level physics prompts. The prompts are mined from physics textbooks via a rigorous processing pipeline. The two stage eval protocol is designed for this benchmark, in which a rule based metric used followed by using llm (gpt4o) to double check those marked as wrong. The authors compare many LLMs on this benchmarks and the best one's score is less than 50 out of 100, so it could be a good benchmark for reasoning models for some time.

**Claims And Evidence:**

This is a new benchmark paper, so i have less concern on this.

**Essential References Not Discussed:**

n/a

**Experimental Designs Or Analyses:**

The experiments are mainly comparing LLMs in the proposed benchmark, which i don't find clear concern.

**Methods And Evaluation Criteria:**

- The method of creating this benchmark is reasonable. All questions are grounded on physics textbooks and the extracted latex format goes through manual check, so the quality should be good.
- Eval for STEM questions is not easy since the gold answer could be freeform (unlike math is more formalized), so the proposed two stage eval protocol is a reasonable approach, though i'd hope there is a "STEM sympy" someday.

**Other Comments Or Suggestions:**

Please check my previous section.

**Other Strengths And Weaknesses:**

Weakness:
 - It would be great if the authors can provide an analysis how robust the MARJ eval method is. e.g., how often it makes wrong judgement? In what scenarios LLMs can't correctly compare the given solution with gold?
-  It would be great if the authors can run some stats on the complexity / difficulty of this benchmark. e.g., for o1-like reasoning models, how many tokens do them need to solve the problem on average?
- I'm curious how does the most frontier model e.g., o3-mini performs on this benchmark, since this basically measures the lifecycle of this benchmakr. (I understand there is lots of overhead to run this, especially if the authors are from academia, so I totally understand if the authors don't give this in rebuttal.)

**Questions For Authors:**

- Regrading "The UGPhysics is sourced from several undergraduate-level physics exercise books." What exercise books are used as data source?
- Why using math-specialized LLM for this physics benchmark?

**Relation To Broader Scientific Literature:**

- This paper mainly targets pushing the frontier of LLM reasoning models.

**Theoretical Claims:**

No theoretical claims in this paper.

---

> ### Author Rebuttal · Authors · 2025-03-28
>
> Dear Reviewer 3YX8,
>
> Thank you for your helpful comments! We will reply to your questions one by one as follows:
>
> > It would be great if the authors can provide an analysis how robust the MARJ eval method is. e.g., how often it makes wrong judgement? In what scenarios LLMs can't correctly compare the given solution with gold?
>
> Thank you for your suggestion! We **have conducted such an analysis in Section 5.2**: “We find that our MARJ evaluation achieves an accuracy of 98% when compared to human annotations.” During our manual inspection, we observed that our MARJ sometimes still fails to correctly evaluate answers that are equivalent in physics but require several steps of conversion.
> For instance, consider the ground-truth answer $RT/\mu$ and the model-generated answer $p/\rho$. While both are physically equivalent ($p/\rho= pV/m = nRT/m = RT/\mu$, using the formula $PV = nRT$ and the definition $μ = m/n$), our MARJ fails to recognize the equivalence due to the need for multi-step conversion.
>
> > It would be great if the authors can run some stats on the complexity / difficulty of this benchmark. e.g., for o1-like reasoning models, how many tokens do them need to solve the problem on average?
>
> Thank you for your comment. In fact, we have analyzed the difficulty of UGPhysics through “physics reasoning skills” in Section 5.1 (Figure 2b). As suggested, we will also **add the stats of the tokens** DeepSeek-Distill-Qwen-32B used to solve the problem on average:
>
> | Dataset    | Avg. Tokens  |
> |-------------|--------|
> | UGPhysics   | 4081   |
> | MATH        | 3079   |
>
> In this table, we also include the average tokens DeepSeek-Distill-Qwen-32B spent to solve MATH [1] for reference.
>
> In addition, the average number of tokens that DeepSeek-R1 spent to solve problems in UGPhysics is 5555.
>
> > I'm curious how does the most frontier model e.g., o3-mini performs on this benchmark, since this basically measures the lifecycle of this benchmakr. (I understand there is lots of overhead to run this, especially if the authors are from academia, so I totally understand if the authors don't give this in rebuttal.)
>
> Thank you for your valuable question and understanding! There is indeed a lot of overhead to evaluate o3-mini. We will **add the results of DeepSeek-R1**, whose performance is catching up with o3-mini high(90.8% vs. 86.9% on MMLU). (DeepSeek-R1 is much cheaper)
>
>
>   |Mec. and Ther. (EN) | Mec. and Ther. (ZH) | Elec. (EN) | Elec. (ZH) | Modern Physics (EN) | Modern Physics (ZH) | Overall (EN) | Overall (ZH) | Average  |
> ---------------------|---------------------|------------|------------|----------------------|---------------------|--------------|--------------|----------|
>  0.5549              | 0.5667              | 0.5450     | 0.4839     | 0.5990               | 0.5729              | 0.5716       | 0.5553       | 0.5634   |
>
> From the table, the overall accuracy is 56.34%, which is higher than o1-mini as expected and there is still much room for improvement.
>
>
> > Regrading "The UGPhysics is sourced from several undergraduate-level physics exercise books." What exercise books are used as data source?
>
> Thank you for your question! There is a risk to reveal the institute of several authors and we give the links to these books at a later stage (if possible).
>
> > Why using math-specialized LLM for this physics benchmark?
>
> As we mentioned in L105-108: “The inclusion of math LLMs aims to assess the extent to which training on specialized math corpus contributes to physics reasoning.”  From experiments, we find that “math-specialized LLMs yield only minor improvements over their general-purpose counterparts in UGPhysics, suggesting the compulsion for more high-quality physics corpora.” (L119 –L122).
> We believe the Reviewer zXs9 gives an interesting discussion about this in the section of "Experimental Designs Or Analyses" in his/her review.
>
> We would like to thank you once again for your useful suggestions to improve the quality of our manuscript. Feel free to discuss if you have any further questions or comments.
>
> Sincerely,
>
> Authors
>
> [1] Hendrycks et al., 2021; Measuring Mathematical Problem Solving with the MATH Dataset.

---

### Official Review · Reviewer_zXs9 · 2025-03-19

**Overall Recommendation:** 4

**Summary:**

This paper introduces UGPhysics, a large‐scale, bilingual benchmark specifically designed for evaluating undergraduate-level physics reasoning with large language models. UGPhysics comprises 5,520 distinct physics problems (11,040 when including both English and Chinese versions) spanning 13 subjects and 59 topics. In addition to the dataset, the paper proposes a novel evaluation framework called Model-Assistant Rule-based Judgment (MARJ) that combines rule‐based precision with model‐based flexibility to assess complex, multi-step physics problem solutions. Extensive experiments across 31 LLMs reveal that even state-of-the-art models, such as OpenAI-o1-mini, achieve only about 50% accuracy, underscoring the challenges posed by physics reasoning compared to math-focused tasks.

**Claims And Evidence:**

The paper provides sufficient empirical evidence of the claims made. The main claim of the paper that physics reasoning abilities of LLMs have not received sufficient attention and as a result LLMs struggle on the task has been well substantiated by the observation that the best performance on the proposed benchmark is 49.8% on the proposed benchmark while several math reasoning benchmarks have been saturated.

**Essential References Not Discussed:**

TheoremQA [2] also contains some physics questions but has not been discussed in the paper.

[2] Chen et al., 2024; TheoremQA: A Theorem Driven Question-Answering Dataset

**Experimental Designs Or Analyses:**

The authors provide an elaborate set of experiments and discussion of their results on the benchmarks. The observation that Math specialised LLMs do not necessarily perform better on Physics as compared to their general counterparts is interesting, showing that finetuning on specific maths problems does not necessarily lead to an improvement in general reasoning capabilities. At the same time, the fact that o1 like models which are post-trained predominantly on Math / Code rasoning data, perform the best suggests that RL based post training can lead to improvements in general reasoning capabilities of models.

**Methods And Evaluation Criteria:**

The methodological description and as described are generally speaking quite sound. Prior works on mathematical reasoning (such as [1]) evaluation of LLMs use evaluation techniques similar to the MARJ evaluation (i.e. using a combination of rule based + LLM-as-a-judge) method described in the paper (without explicitly describing the procedure). Regardless, I believe that stating the use of, and describing the procedure explicitly is a valuable contribution.

[1] Didolkar et al., 2024;  Metacognitive Capabilities of LLMs: An Exploration in Mathematical Problem Solving

**Other Comments Or Suggestions:**

Including a comparison of performances of models with some standard physics reasoning benchmarks (such as MMLU Physics subset, PhysicsQA, etc. to that on UGPhysics would help give the reader a better idea of the overall difficulty of the benchmark as compared to existing benchmarks.

**Other Strengths And Weaknesses:**

All strengths and weaknesses of the paper have been discussed in other sections.

**Questions For Authors:**

The authors mention that the initial questions are in Chinese and are then translated to English. How is this translation done? If it is done using LLMs / some other machine translation methods, are any measures undertaken in order to ensure a high quality of the translations?

**Relation To Broader Scientific Literature:**

This paper falls within the vast literature on LLM evaluation - specifically evaluating Physics reasoning capabilities of LLMs. While there exist multiple evalution benchmarks for phsics, most of them are either simple for existing LLMs, do not require elaborate CoTs, are limited in size or do not cover a wide range of topics. This value of this work stems from its elaborate subject categorization, support for two languages, difficulty level, larger size and an elaborate evaluation pipeline.

**Theoretical Claims:**

The paper does not make any theoretical claims

---

> ### Author Rebuttal · Authors · 2025-03-27
>
> Dear Reviewer zXs9,
>
> Thank you for your constructive feedback! We will reply to your questions one by one as follows:
>
> > Prior works on mathematical reasoning (such as [1]) evaluation of LLMs use evaluation techniques similar to the MARJ evaluation (i.e. using a combination of rule based + LLM-as-a-judge) method described in the paper (without explicitly describing the procedure). Regardless, I believe that stating the use of, and describing the procedure explicitly is a valuable contribution.
>
> Thank you for pointing out this relevant paper and acknowledging our contribution.
>
> After reading [1], they employ a model-based evaluation to get additional metrics from three angles, which differs slightly from our setting.
>
> We will **include the following sentence in the "Related Work" section** (in "Answer Judgment") for discussion:
>
> "Additionally, several works [1, 2] utilize model-based evaluation to obtain additional metrics for assessing effectiveness."
>
>
> > TheoremQA also contains some physics questions but has not been discussed in the paper.
>
> Thank you for pointing this out. We will **add the following line to Table 1**:
>
> | Dataset     | Level | # Test | # UG | Subjects | # Ans. Types | Language | Eval. | Leak. Det. |
> |-------------|-------|--------|------|----------|--------------|----------|-------|------------|
> | TheoremQA   | 5     | 131    | 131  | –        | 5            | EN       | Rule  | No         |
>
> > Including a comparison of performances of models with some standard physics reasoning benchmarks (such as MMLU Physics subset, PhysicsQA, etc. to that on UGPhysics would help give the reader a better idea of the overall difficulty of the benchmark as compared to existing benchmarks.
>
> Thank you for your suggestion! We will **add this comparison for the GPT-4o model as follows** (we also include MATH for reference) and will include a figure to illustrate this table in our manuscript as well:
>
> | Dataset          | Performance       |
> |------------------|-------------------|
> | Ours             | 38.67%           |
> | MMLU   (college physics)          | 68.6%   |
> |       MMLU  (high school physics)          | 72.8%  |
> |      MMLU  ((conceptual physics) )          | 92.3%  |
> | MMLU-pro         | 75.06%           |
> | OlympicArena     | 55.92%           |
> | GPQA             | 53.6%            |
> | MATH             | 76.6%            |
>
>
>
>
> > The authors mention that the initial questions are in Chinese and are then translated to English. How is this translation done? If it is done using LLMs / some other machine translation methods, are any measures undertaken in order to ensure a high quality of the translations?
>
> Following [3, 4], we leverage LLMs (specifically GPT-4o-2024-08-06) for translation. As demonstrated in [3] (using GPT-4 for translation) and [4] (using GPT-3.5-turbo), the quality of translation produced by LLMs is high. Since we utilize a significantly more powerful model, the translation quality is expected to be even higher.
>
> Furthermore, during the initial stages of translation, we manually reviewed several examples (typically 5-20) for each subject (particularly checking whether it can handle physics-specialized terminology). This process confirmed that GPT-4o excels at translating them.
>
>
> Thank you once again for your valuable suggestions to improve the quality of our work. If you have any further questions or feedback, please do not hesitate to reach out to us.
>
> Sincerely,
>
> Authors
>
> [1] Didolkar et al., 2024; Metacognitive Capabilities of LLMs: An Exploration in Mathematical Problem Solving
>
> [2] Huang et al., 2024; Olympicarena: Benchmarking Multi-Discipline Cognitive Reasoning for Superintelligent AI.
>
> [3] Liu et al., 2024; Mathbench: Evaluating the Theory and Application Proficiency of LLMs with a Hierarchical Mathematics Benchmark.
>
> [4] Tang et al., 2024; Mathscale: Scaling Instruction Tuning for Mathematical Reasoning.

---

> > ### Comment · Reviewer_zXs9 · 2025-04-03
> >
> > Thank you for the reply and clarifications. I would like to maintain my score.

---

### Decision · Program_Chairs · 2025-05-01

**Decision:**

Accept (poster)

**Comment:**

The paper introduces UGPhysics, a large-scale benchmark of 5,520 undergraduate-level physics problems in English and Chinese, designed to comprehensively evaluate LLMs across 13 subjects and multiple reasoning skills. Using a novel Model-Assistant Rule-based Judgment (MARJ) pipeline, the authors assess 31 LLMs and find that even top models achieve only 49.8% accuracy, highlighting the need for improved physics reasoning capabilities in LLMs.

Overall the benchmark seems useful and helpful, and it addresses a timely problem. All reviewers are positive and the concerns of the less positive reviews seem to be overall answered in the discussion.